# Improved motif-scaffolding with SE(3) flow matching

**Jason Yim**                                                                              *jyim@csail.mit.edu*
*Computer Science and Artificial Intelligence Laboratory*
*Massachusetts Institute of Technology*

**Andrew Campbell**                                                                *campbell@stats.ox.ac.uk*
*Department of Statistics*
*University of Oxford*

**Emile Mathieu**                                                                          *ebm32@cam.ac.uk*
*Department of Engineering*
*University of Cambridge*

**Andrew Y. K. Foong**                                                        *andrewfoong@microsoft.com*
*Microsoft Research AI4Science*

**Michael Gastegger**                                                          *mgastegger@microsoft.com*
*Microsoft Research AI4Science*

**José Jiménez-Luna**                                                        *jjimenezluna@microsoft.com*
*Microsoft Research AI4Science*

**Sarah Lewis**                                                                    *sarahlewis@microsoft.com*
*Microsoft Research AI4Science*

**Victor Garcia Satorras**                                                        *victorgar@microsoft.com*
*Microsoft Research AI4Science*

**Bastiaan S. Veeling**                                                            *basveeling@microsoft.com*
*Microsoft Research AI4Science*

**Frank Noé**                                                                        *franknoe@microsoft.com*
*Microsoft Research AI4Science*

**Regina Barzilay**                                                                      *regina@csail.mit.edu*
*Computer Science and Articial Intelligence Laboratory*
*Massachusetts Institute of Technology*

**Tommi S. Jaakkola**                                                                *tommi@csail.mit.edu*
*Computer Science and Articial Intelligence Laboratory*
*Massachusetts Institute of Technology*

**Reviewed on OpenReview:** *https://openreview.net/forum?id=fa1ne8xDGn*

## Abstract

Protein design often begins with the knowledge of a desired function from a motif which motif-scaffolding aims to construct a functional protein around. Recently, generative models have achieved breakthrough success in designing scaffolds for a range of motifs. However, generated scaffolds tend to lack structural diversity, which can hinder success in wet-lab validation. In this work, we extend FrameFlow, an SE(3) flow matching model for protein backbone generation, to perform motif-scaffolding with two complementary approaches. The first is *motif amortization*, in which FrameFlow is trained with the motif as input using a data augmentation strategy. The second is *motif guidance*, which performs scaffolding using an estimate of the conditional score from FrameFlow without additional training. On a benchmark of 24 biologically meaningful motifs, we show our method achieves 2.5 times more designable and unique motif-scaffolds compared to state-of-the-art. Code: https://github.com/microsoft/protein-frame-flow

# 1 Introduction

A common task in protein design is to create proteins with functional properties conferred through a pre-specified arrangement of residues known as a *motif*. The problem is to design the remainder of the protein, called the *scaffold*, that harbors the motif. Motif-scaffolding is widely used, with applications to vaccine and enzyme design (Procko et al., 2014; Correia et al., 2014; Jiang et al., 2008; Siegel et al., 2010). For this problem, diffusion models have greatly advanced capabilities in designing new scaffolds (Wu et al., 2023; Trippe et al., 2022; Ingraham et al., 2023). While experimental wet-lab validation is the ultimate test for evaluating a scaffold, in this work we focus on improving performance under computational validation of scaffolds following prior works. *In-silico* success is defined as satisfying the designability[1] criteria which has been found to correlate well with wet-lab success (Wang et al., 2021). The current state-of-the-art, RFdiffusion (Watson et al., 2023), fine-tunes a pre-trained RosettaFold (Baek et al., 2023) neural network with SE(3) diffusion (Yim et al., 2023b) and is able to successfully scaffold the majority of motifs in a recent benchmark.[2] However, RFdiffusion suffers from low scaffold diversity which can hinder chances of a successful design. Moreover, the large model size and pre-training used in RFdiffusion makes it slow to train and difficult to deploy on smaller machines. In this work, we present a lightweight and easy-to-train model with improved performance.

Our method adapts an existing SE(3) flow matching model, FrameFlow (Yim et al., 2023a), for motif-scaffolding. We develop two approaches: (i) *motif amortization*, and (ii) *motif guidance* as illustrated in Fig. 1. Motif amortization simply trains a *conditional* model with the motif as additional input when generating the scaffold. We use data augmentation to amortize over all possible motifs in our training set and aid in generalization to new motifs. Motif guidance relies on a Bayesian approach, using an *unconditional* FrameFlow model to sample the scaffold residues, while the motif residues are guided at each step to their final desired positions. An unconditional model in this context is one that generates the full protein backbone without distinguishing between the motif and scaffold. Motif guidance was described in Wu et al. (2023) for SE(3) diffusion. In this work, we develop the extension to SE(3) flow matching.

The two approaches differ in whether to use an conditional model or to re-purpose an unconditional model for conditional generation. Motif guidance has the advantage that any unconditional model can be used to readily perform motif scaffolding without the need for additional task-specific training. To provide a controlled comparison, we train unconditional and conditional versions of FrameFlow on a dataset of monomers from the Protein Data Bank (PDB) (Berman et al., 2000). Our results provide a clear comparison of the modeling choices made when performing motif-scaffolding with FrameFlow. We find that FrameFlow with

---

[1] A metric based on using ProteinMPNN (Dauparas et al., 2022) and AlphaFold2 (Jumper et al., 2021) to determine the quality of a protein backbone.

[2] First introduced in RFdiffusion as a benchmark of 24 single-chain motifs successfully solved across prior works published.

Figure 1: We present two strategies for motif-scaffolding. **Top**: motif amortization trains a flow model to condition on the motif (blue) and generate the scaffold (red). During training, only the scaffold is corrupted with noise. **Bottom**: motif guidance re-purposes a flow model that is trained to generate the full protein for motif-scaffolding. During generation, the motif residues are guided to reconstruct the true motif at $t = 1$ while the flow model will adjust the scaffold trajectory to be consistent with the motif.

both motif amortization and guidance surpasses the performance of RFdiffusion, as measured by the number of structurally *unique* scaffolds[3] that pass the designability criterion.

This work is structured as follows. Sec. 2 provides background on SE(3) flow matching. We present our main contribution extending FrameFlow for motif-scaffolding in Sec. 3. We develop motif amortization for flow matching while motif guidance, originally developed for diffusion models, follows after drawing connections between flow matching and diffusion models. Next we discuss related works Sec. 4 and present empirical results Sec. 5. Our contributions are the following:

- We extend FrameFlow with two fundamentally different approaches for motif-scaffolding: motif amortization and motif guidance. We are the first to extend conditional generation techniques with SE(3) flow matching and apply them to motif-scaffolding. With all other settings kept constant, we perform a empirical study of how each approach performs.

- On a benchmark of biologically meaningful motifs, we show our method can successfully scaffold 20 out of 24 motifs in the motif-scaffolding benchmark which is equivalent to previous state-of-the-art, while achieving 2.5 times more unique, designable scaffolds. Our results demonstrate the importance of measuring diversity to detect mode collapse.

## 2 Background

Flow matching (FM) (Lipman et al., 2023; Albergo et al., 2023) is a simulation-free method for training continuous normalizing flows (CNFs) (Chen et al., 2018). CNFs are deep generative models that generates data by integrating an ordinary differential equation (ODE) over a learned vector field. Recently, flow matching has been extended to Riemannian manifolds (Chen & Lipman, 2023), which we rely on to model protein backbones via the local frame SE(3) representation. Sec. 2.1 gives an introduction to Riemannian flow matching. Sec. 2.2 then describes how SE(3) flow matching is applied to protein backbones.

### 2.1 Flow matching on Riemannian manifolds

On a manifold $\mathcal{M}$, a CNF $\phi_t(\cdot) : \mathcal{M} \to \mathcal{M}$ is defined via an ODE along a time-dependent vector field $v(z, t) : \mathcal{M} \times \mathbb{R} \to \mathcal{T}_z\mathcal{M}$ where $\mathcal{T}_z\mathcal{M}$ is the tangent space of the manifold at $z \in \mathcal{M}$ and time is $t \in [0, 1]$:

$$\frac{\mathrm{d}}{\mathrm{d}t}\phi_t(z_0) = v(\phi_t(z_0), t), \ \phi_0(z_0) = z_0. \tag{1}$$

---

[3]The number of unique scaffolds is defined as the number of structural clusters. See Sec. 5.1

Starting with $z_0 \sim p_0$ from an easy-to-sample prior distribution $p_0$, simulating samples according to Eq. (1) induces a new distribution referred as the push-forward $p_t = [\phi_t]_* p_0$. One wishes to find a vector field $v$ such that the push-forward $p_{t=1} = [\phi_{t=1}]_* p_0$ (at $t = 1$) matches the data distribution $p_1$. Such a vector field $v$ is in general not available in closed-form, but can be learned by regressing conditional vector fields $u(z_t, t|z_1) = \frac{d}{dt} z_t$ where $z_t = \phi_t(z_0|z_1)$ interpolates between endpoints $z_0 \sim p_0$ and $z_1 \sim p_1$. A natural choice for $z_t$ is the geodesic path: $z_t = \exp_{z_0}\left(t \log_{z_0}(z_1)\right)$, where $\exp_{z_0}$ and $\log_{z_0}$ are the exponential and logarithmic maps at the point $z_0$. The conditional vector field takes the following form: $u(z_t, t|z_1) = \log_{z_t}(z_1)/(1-t)$. The key insight of conditional[4] flow matching (CFM) (Lipman et al., 2023) is that training a neural network $\hat{v}$ to regress the conditional vector field $u$ is equivalent to learning the unconditional vector field $v$. This corresponds to minimizing

$$\mathcal{L} = \mathbb{E}_{\mathcal{U}(t;[0,1]), p_1(z_1), p_0(z_0)} \left[ \|u(z_t, t|z_1) - \hat{v}(z_t, t)\|_g^2 \right] \tag{2}$$

where $\mathcal{U}(t; [0, 1])$ is the uniform distribution for $t \in [0, 1]$ and $\|\cdot\|_g^2$ is the norm induced by the Riemannian metric $g : \mathcal{TM} \times \mathcal{TM} \to \mathbb{R}$. Samples can then be generated by integrating the ODE in Eq. (1) with Euler steps using the learned vector field $\hat{v}$ in place of $v$.

## 2.2 Generative modeling on protein backbones

The atom positions of each residue in a protein backbone can be parameterized by an element $T \in \mathrm{SE}(3)$ of the special Euclidean group $\mathrm{SE}(3)$ (Jumper et al., 2021; Yim et al., 2023b). We refer to $T = (r, x)$ as a (local) frame consisting of a rotation $r \in \mathrm{SO}(3)$ and translation vector $x \in \mathbb{R}^3$. The protein backbone is made of $N$ residues, meaning it can be parameterized by $N$ frames denoted as $\mathbf{T} = [T^{(1)}, \ldots, T^{(N)}] \in \mathrm{SE}(3)^N$. We use bold face to refer to vectors of all the residues, superscripts to refer to residue indices, and subscripts refer to time. Details of the $\mathrm{SE}(3)^N$ backbone parameterization can be found in App. B.1.

We use $\mathrm{SE}(3)$ flow matching to parameterize a generative model over the $\mathrm{SE}(3)^N$ representation of protein backbones. The application of Riemannian flow matching to $\mathrm{SE}(3)$ was previously developed in Yim et al. (2023a); Bose et al. (2023). Endowing $\mathrm{SE}(3)$ with the product left-invariant metric, the $\mathrm{SE}(3)$ manifold effectively behaves as the product manifold $\mathrm{SE}(3) = \mathrm{SO}(3) \times \mathbb{R}^3$ (App. D.3 of Yim et al. (2023b)). The vector field over $\mathrm{SE}(3)$ can then be decomposed as $v_{\mathrm{SE}(3)}^{(n)}(\cdot, t) = (v_{\mathbb{R}}^{(n)}(\cdot, t), v_{\mathrm{SO}(3)}^{(n)}(\cdot, t))$. Our goal is train a neural network to parameterize the learned vector fields,

$$\hat{v}_{\mathbb{R}}^{(n)}(\mathbf{T}_t, t) = \frac{\hat{x}_1^{(n)}(\mathbf{T}_t) - x_t^{(n)}}{1 - t}, \qquad \hat{v}_{\mathrm{SO}(3)}^{(n)}(\mathbf{T}_t, t) = \frac{\log_{r_t^{(n)}}(\hat{r}_1^{(n)}(\mathbf{T}_t))}{1 - t}. \tag{3}$$

The outputs of the neural network are *denoised* predictions $\hat{x}_1^{(n)}$ and $\hat{r}_1^{(n)}$ which are used to calculate the vector fields in Eq. (3). The loss becomes

$$\mathcal{L}_{\mathrm{SE}(3)} = \mathbb{E}\left[ \left\| \mathbf{u}_{\mathrm{SE}(3)}(\mathbf{T}_t, t|\mathbf{T}_1) - \hat{\mathbf{v}}_{\mathrm{SE}(3)}(\mathbf{T}_t, t) \right\|_{\mathrm{SE}(3)}^2 \right] \tag{4}$$

$$= \mathbb{E}\left[ \left\| \mathbf{u}_{\mathbb{R}}(\mathbf{x}_t, t|\mathbf{x}_1) - \hat{\mathbf{v}}_{\mathbb{R}}(\mathbf{T}_t, t) \right\|_{\mathbb{R}}^2 + \left\| \mathbf{u}_{\mathrm{SO}(3)}(\mathbf{r}_t, t|\mathbf{r}_1) - \hat{\mathbf{v}}_{\mathrm{SO}(3)}(\mathbf{T}_t, t) \right\|_{\mathrm{SO}(3)}^2 \right] \tag{5}$$

where the expectation is taken over $\mathcal{U}(t; [0, 1])$, $p_1(\mathbf{T}_1)$, $p_0(\mathbf{T}_0)$. We have used bold-face for collections of elements, i.e. $\hat{\mathbf{v}}(\cdot) = [\hat{v}^{(1)}(\cdot), \ldots, \hat{v}^{(N)}(\cdot)]$. Our prior is chosen as $p_0(\mathbf{T}_0) = \mathcal{U}(\mathrm{SO}(3))^N \otimes \overline{\mathcal{N}}(0, I_3)^N$, where $\mathcal{U}(\mathrm{SO}(3))$ is the uniform distribution over $\mathrm{SO}(3)$ and $\overline{\mathcal{N}}(0, I_3)$ is the isotropic Gaussian where samples are centered to the origin. Details of $\mathrm{SE}(3)$ flow matching such as architecture and hyperparameters closely follow FrameFlow (Yim et al., 2023a), details of which are provided in App. B.2.

## 3 Motif-scaffolding with FrameFlow

We describe our two strategies for performing motif-scaffolding with the FrameFlow model: motif amortization (Sec. 3.1) and motif guidance (Sec. 3.2). Recall the full protein backbone is given by $\mathbf{T} =$

---

[4]Unfortunately the meaning of "conditional" is overloaded. The conditionals will be clear from the context.

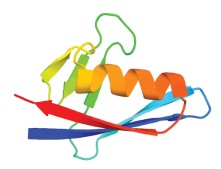 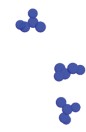 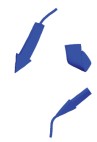 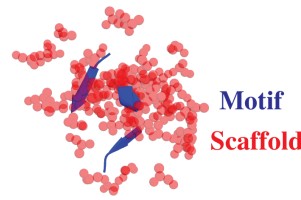

**Motif**
**Scaffold**

**1. Sample protein from dataset**  **2. Sample number and location of motif(s).**  **3. Sample motif lengths.**  **4. Corrupt remaining residues (scaffold).**

Figure 2: **Motif data augmentation.** Each protein in the dataset does not come with pre-defined motif-scaffold annotations. Instead, we construct plausible motifs at random to simulate sampling from the distribution of motifs and scaffolds.

$\{T^{(1)}, T^{(2)}, \ldots, T^{(N)}\} \in \mathrm{SE}(3)^N$. The residues can be separated into the motif $\mathbf{T}^M = \{T^{(i_1)}, \ldots, T^{(i_k)}\}$ of length $k$ where $\{i_1, \ldots, i_k\} \subset \{1, \ldots, N\}$ are motif residue indices, and the scaffold $\mathbf{T}^S$ is all the remaining residues, such that $\mathbf{T} = \mathbf{T}^M \cup \mathbf{T}^S$. The task can then be framed as the problem of sampling from the conditional distribution $p(\mathbf{T}^S | \mathbf{T}^M)$.

### 3.1 Motif amortization

We train a variant of FrameFlow that additionally takes the motif as input when generating scaffolds (and keeping the motif fixed). Formally, we model a motif-conditioned CNF via the following ODE,

$$\frac{\mathrm{d}}{\mathrm{d}t}\phi_t(\mathbf{T}_0^S | \mathbf{T}^M) = v(\phi_t, t | \mathbf{T}^M), \quad \phi_0(\mathbf{T}_0^S | \mathbf{T}^M) = \mathbf{T}_0^S. \tag{6}$$

The flow $\phi_t$ transforms a prior density over scaffolds along time, inducing a density $p_t(\cdot | \mathbf{T}^M) = [\phi_t]_* p_0(\cdot | \mathbf{T}^M)$. We use the same prior as in Sec. 2.2: $p_0(\mathbf{T}_0^S | \mathbf{T}^M) = p_0(\mathbf{T}_0^S)$. FrameFlow is trained to predict the conditional vector field $u(\mathbf{T}_t^S, t | \mathbf{T}_1^S, \mathbf{T}^M)$ where $\mathbf{T}_t^S$ is defined by interpolating along the geodesic path, $\mathbf{T}_t^S = \exp_{\mathbf{T}_0^S}\left(t \log_{\mathbf{T}_0^S}(\mathbf{T}_1^S)\right)$. The implication is that $u$ is conditionally independent of the motif $\mathbf{T}^M$ given $\mathbf{T}_1^S$. This simplifies our formulation to $u(\mathbf{T}_t^S, t | \mathbf{T}_1^S, \mathbf{T}^M) = u(\mathbf{T}_t^S, t | \mathbf{T}_1^S)$ that is defined in Sec. 2.2. However, when we learn the vector field, the model needs to condition on $\mathbf{T}^M$ since the motif placement $\mathbf{T}^M$ contains information on the true scaffold positions $\mathbf{T}_1^S$. The training loss becomes,

$$\mathbb{E}\left[\left\|\mathbf{u}_{\mathrm{SE}(3)}(\mathbf{T}_t^S, t | \mathbf{T}_1^S) - \hat{\mathbf{v}}_{\mathrm{SE}(3)}(\mathbf{T}_t^S, t | \mathbf{T}^M)\right\|_{\mathrm{SE}(3)}^2\right] \tag{7}$$

where the expectation is taken over $\mathcal{U}(t; [0, 1])$, $p(\mathbf{T}^M)$, $p_1(\mathbf{T}_1^S | \mathbf{T}^M)$, $p_0(\mathbf{T}_0^S)$. The above expectation requires access to the motif and scaffold distributions, $p(\mathbf{T}^M)$ and $p_1(\mathbf{T}_1^S | \mathbf{T}^M)$, during training. Future work can look into incorporating known motif-scaffolds such as the CDR loops on antibodies (Dunbar et al., 2014). While some labels exist for which residues correspond to the functional motif, the vast majority of protein structures in the PDB do not have labels. We instead utilize unlabeled PDB structures to perform data augmentation (see Sec. 3.1.1) that allows sampling a wide range of motifs and scaffolds.

To learn the motif-conditioned vector field $\hat{v}_t$, we use the FrameFlow architecture with a 1D mask as additional input with a 1 at the location of the motif and 0 elsewhere. To maintain SE(3)-equivariance, we zero-center the motif and initial noise sample from $p_0(\mathbf{T}_0^S | \mathbf{T}^M)$. Zero-centering the motif also prevents the model from using the motif offset from the origin to memorize scaffold locations which helps generalization.

### 3.1.1 Data augmentation.

The flow matching loss from Eq. (7) involves sampling from $p(\mathbf{T}^M)$ and $p_1(\mathbf{T}_1^S | \mathbf{T}^M)$, which we do not have access to, but can be approximated using unlabeled structures from the PDB. Our pseudo-labeled motifs and scaffolds are generated as follows (also depicted in Fig. 2). First, a protein structure is sampled from the PDB dataset. Second, a random number of residues are selected to be the starting locations of each motif. Third, additional residues are appended onto each motif thereby extending their lengths. The length of each motif is randomly sampled such that the total number of motif residues is between $\gamma_{\min}$ and $\gamma_{\max}$ percent of all the residues. We use $\gamma_{\min} = 0.05$ and $\gamma_{\max} = 0.5$ to ensure at least a few residues are used as the motif

but not more than half the protein. Finally, the remaining residues are treated as the scaffold and corrupted. The motif and scaffold are treated as samples from $p(\mathbf{T}^M)$ and $p_1(\mathbf{T}_1^S|\mathbf{T}^M)$ respectively. Importantly, each protein will be re-used on subsequent epochs where new motifs and scaffolds will be sampled. Our pseudo motif-scaffolds cover a wide range of scenarios that cover multiple motifs of different lengths.

The lack of functional annotations in the PDB requires training over all possible motif-scaffold annotations to handle new scenarios our method may encounter in real world scenarios. In our experiments, we evaluate how this data augmentation strategy transfers to real motif-scaffolding tasks. A similar strategy is used in image infilling where image based diffusion models are trained to infill randomly masked crops of images to approximate real image infilling scenarios (Saharia et al., 2022). Motif-scaffolding data augmentation was mentioned in RFdiffusion but without algorithmic detail. Since RFdiffusion does not release training code, we implemented our own data augmentation algorithm in Algorithm 1.

## 3.2 Motif guidance

We now present an alternative Bayesian approach to motif-scaffolding that does not involve learning a motif-conditioned flow model. As such it does not require having access to motifs at training time, but only at sampling time. This can be useful when an unconditional generative flow model is already available at hand and additional training is too costly. The idea behind motif guidance, first described as a special case of TDS (Wu et al., 2023) using diffusion models, is to use the desired motif $\mathbf{T}^M$ to bias the model's generative trajectory such that the motif residues end up in their known positions. The scaffold residues follow a trajectory that create a consistent whole protein backbone, thus achieving motif-scaffolding.

The key insight comes from connecting flow matching to diffusion models to which motif guidance can be applied. The following ODE describes the relationship between the vector field $\hat{\mathbf{v}}$ in flow models – learned by minimizing CFM objective in Eq. (5) – and the Stein score $\nabla \log p_t(\mathbf{T}_t)$,

$$\mathrm{d}\mathbf{T}_t = \hat{\mathbf{v}}(\mathbf{T}_t, t)\mathrm{d}t = \left[f(\mathbf{T}_t, t) - \frac{1}{2}g(t)^2 \nabla \log p_t(\mathbf{T}_t)\right]\mathrm{d}t. \tag{8}$$

The gradient is taken with respect to the backbone at time $t$ which we omit for brevity, i.e. $\nabla = \nabla_{\mathbf{T}_t}$. Eq. (8) shows the ODE used to sample from flow models can be written as the probability flow ODE used in diffusion models (Song et al., 2020) with $f$ and $g$ as the drift and diffusion coefficients. The derivation of Eq. (8) requires standard linear algebra and calculus for our choice of vector field (see App. D).

Our goal is to sample from the conditional $p(\mathbf{T}|\mathbf{T}^M)$ from which we can extract $p(\mathbf{T}^S|\mathbf{T}^M)$. The benefit of Eq. (8) is we can manipulate the score term to achieve this goal. We modify the above to be conditioned on the motif $\mathbf{T}^M$ followed by an application of Bayes rule where $\nabla \log p_t(\mathbf{T}_t|\mathbf{T}^M) = \nabla \log p_t(\mathbf{T}_t) + \nabla \log p_t(\mathbf{T}^M|\mathbf{T}_t)$.

$$\mathrm{d}\mathbf{T}_t = \left[f(\mathbf{T}_t, t) - \frac{1}{2}g(t)^2 \nabla \log p_t(\mathbf{T}_t|\mathbf{T}^M)\right]\mathrm{d}t \tag{9}$$

$$= \left[f(\mathbf{T}_t, t) - \frac{1}{2}g(t)^2 \left(\nabla \log p_t(\mathbf{T}_t) + \nabla \log p_t(\mathbf{T}^M|\mathbf{T}_t)\right)\right]\mathrm{d}t$$

$$= \left[\underbrace{\hat{\mathbf{v}}_{\mathrm{SE}(3)}(\mathbf{T}_t, t)}_{\text{unconditional pred.}} - \frac{1}{2}g(t)^2 \underbrace{\nabla \log p_t(\mathbf{T}^M|\mathbf{T}_t)}_{\text{guidance term}}\right]\mathrm{d}t. \tag{10}$$

We can interpret Eq. (9) as doing unconditional generation by following $\hat{\mathbf{v}}_{\mathrm{SE}(3)}(\mathbf{T}, t)$ while $\nabla \log p_t(\mathbf{T}^M|\mathbf{T}_t)$ guides the noised residues so as to be consistent with the true motif. Doob's H-transform ensures Eq. (9) will sample from $p(\mathbf{T}|\mathbf{T}^M)$ (Didi et al., 2023). The conditional score $\nabla \log p_t(\mathbf{T}^M|\mathbf{T}_t)$ is unknown, yet it can be approximated by marginalising out $\mathbf{T}_1$ and using the neural network's denoised output (Song et al., 2022; Chung et al., 2022; Wu et al., 2023),

$$p_t(\mathbf{T}^M|\mathbf{T}_t) = \int p(\mathbf{T}^M|\mathbf{T}_1)p_{1|t}(\mathbf{T}_1|\mathbf{T}_t)\mathrm{d}\mathbf{T}_1 \tag{11}$$

$$\approx \int p(\mathbf{T}^M|\mathbf{T}_1)\delta_{\hat{\mathbf{T}}_1^M(\mathbf{T}_t)}(\mathbf{T}_t)\mathrm{d}\mathbf{T}_1 = p(\mathbf{T}^M|\hat{\mathbf{T}}_1^M(\mathbf{T}_t)). \tag{12}$$

We now have the choice to define the likelihood in Eq. (12) to have higher probability the closer it is to the desired motif:

$$p(\mathbf{T}^M|\hat{\mathbf{T}}_1^M(\mathbf{T}_t)) \propto \exp\left(-\|\mathbf{x}^M - \hat{\mathbf{x}}_1^M(\mathbf{T}_t)\|_{\mathbb{R}}^2/\omega_t^2\right)\exp\left(-\|\mathbf{r}^M - \hat{\mathbf{r}}_1^M(\mathbf{T}_t)\|_{\mathrm{SO}(3)}^2/\omega_t^2\right), \tag{13}$$

which is inversely proportional to the distance from the desired motif. Following SE(3) flow matching, Eq. (9) becomes factorized into the translation and rotation components. Plugging $p(\mathbf{T}^M|\hat{\mathbf{T}}_1^M(\mathbf{T}_t))$ in Eq. (9), we arrive at the following ODE we may sample $p(\mathbf{T}|\mathbf{T}^M)$ from

$$\text{Translations: } \mathrm{d}\mathbf{x}_t = \left[\hat{\mathbf{v}}_{\mathbb{R}}(\mathbf{T}_t, t) + \tfrac{1}{2}g(t)^2\nabla_{\mathbf{x}_t}\|\mathbf{x}^M - \hat{\mathbf{x}}_1^M(\mathbf{T}_t)\|_{\mathbb{R}}^2/\omega_t^2\right]\mathrm{d}t. \tag{14}$$

$$\text{Rotations: } \mathrm{d}\mathbf{r}_t = \left[\hat{\mathbf{v}}_{\mathrm{SO}(3)}(\mathbf{T}_t, t) + \tfrac{1}{2}g(t)^2\nabla_{\mathbf{r}_t}\|\mathbf{r}^M - \hat{\mathbf{r}}_1^M(\mathbf{T}_t)\|_{\mathrm{SO}(3)}^2/\omega_t^2\right]\mathrm{d}t. \tag{15}$$

$\omega_t$ is a hyperparameter that controls the magnitude of the guidance towards the desired motif which we set to $\omega_t^2 = (1-t)^2/(t^2 + (1-t)^2)$ as done in Pokle et al. (2023); Song et al. (2021). While different choices of $g(t)$ are possible, Pokle et al. (2023) proposed to use $g(t) = (1-t)/t$ with the motivation that this matches the diffusion coefficient for the diffusion SDE that matches the marginals of the flow ODE. For completeness, we provide the proof for $g(t)$ in App. D. A similar calculation is non-trivial for SO(3), hence we use the same $g(t)$ as a reasonnable heuristic and observe good performance as done in (Wu et al., 2023).

## 4 Related work

**Conditional diffusion and flows.** The development of conditional generation methods for diffusion and flow models is an active area of research. Two popular diffusion techniques that have been extended to flow matching are classifier-free guidance (CFG) (Dao et al., 2023; Ho & Salimans, 2022; Zheng et al., 2023) and reconstruction guidance (Pokle et al., 2023; Ho et al., 2022; Song et al., 2022; Chung et al., 2022). Motif guidance is an application of reconstruction guidance for motif-scaffolding. Motif amortization is most related to data-dependent couplings (Albergo et al., 2023), where a flow is learned with conditioning of partial data.

**Motif-scaffolding.** Wang et al. (2021) first formulated motif-scaffolding using deep learning. SMCDiff (Trippe et al., 2022) was the first proposed diffusion model for motif-scaffolding using Sequential Monte Carlo (SMC). Twisted Diffusion Sampler (TDS) (Wu et al., 2023) later improved upon SMCDiff using reconstruction guidance for each particle in SMC. Our motif guidance method follows from TDS (with one particle) by deriving the equivalent guidance vector field from its conditional score counterpart. RFdiffusion (Watson et al., 2023) fine-tunes a pre-trained neural network with motif-conditioned diffusion training. Our FrameFlow-amortization approach in principle follows RFdiffusion's diffusion training, but differs in (i) using flow matching, (ii) not relying expensive pre-training, and (iii) uses a $3\times$ smaller neural network[5]. Didi et al. (2023) provides a survey of structure-based motif-scaffolding methods while proposing Doob's h-transform for motifs-scaffolding. EvoDiff (Alamdari et al., 2023) differs in using a sequence-based diffusion model that performs motif-scaffolding with language model-style masked generation but performance falls short of RFdiffusion and TDS.

## 5 Experiments

In this section, we report the results of training FrameFlow for motif-scaffolding. Sec. 5.1 describes training, sampling, and metrics. Our main results on motif-scafolding are reported in Sec. 5.2 on the benchmark introduced in RFdiffusion. Additional motif-scaffolding analysis is provided in App. G.

### 5.1 Set-up

**Training.** We train two FrameFlow models. FrameFlow-amortization is trained with motif amortization as described in Sec. 3.1 with data augmentation using hyperparameters: $\gamma_{\min} = 0.05$ so the motif is never

---

[5]FrameFlow uses 16.8 million parameters compared to RFdiffusion's 59.8 million.

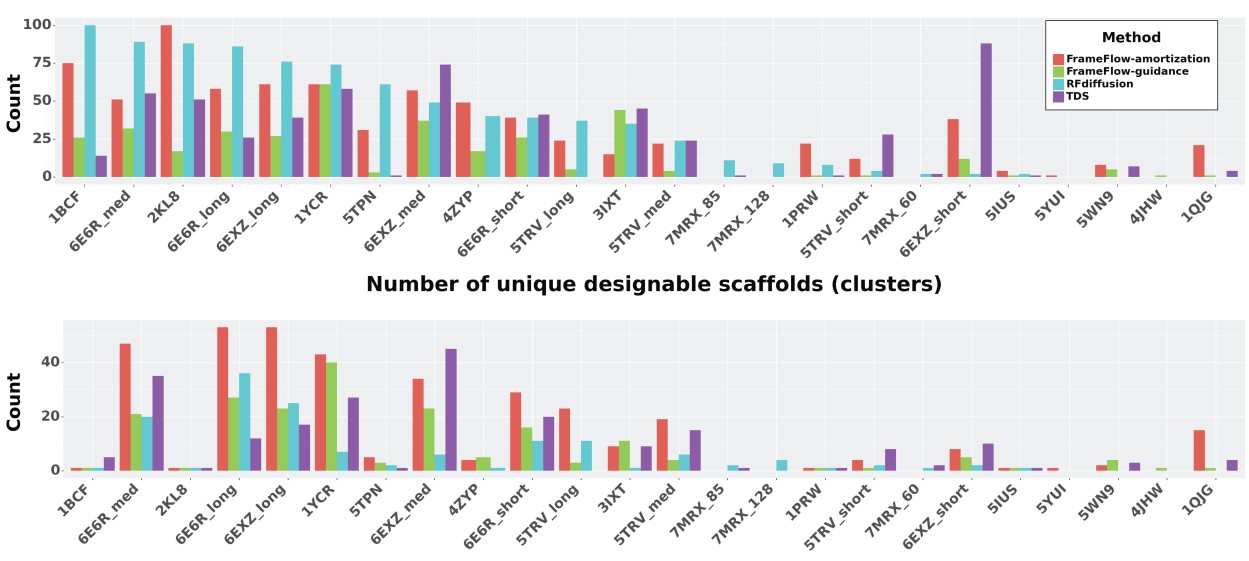

Figure 3: **Motif-scaffolding results.** Top plot: RFdiffusion achieves the most designable scaffolds amongst all methods in 9/24 test motifs compared to FrameFlow-amortization's 7/24 and TDS' 6/24; 2/24 are ties. Bottom plot: However, we observe that RFdiffusion produces the highest number of unique designable scaffolds for only 2 out of the 24 test motifs. Therefore, previous approaches that only measure designability (top plot) may be misleading since those generative models that may have the best designability can also be repeatedly sampling similar scaffolds. This demonstrates the need to measure diversity alongside designability and use the number of unique designable scaffolds as the metric of success.

degenerately small and $\gamma_{\max} = 0.5$ to avoid motif being the majority of the backbone. FrameFlow-guidance, to be used in motif guidance, is trained unconditionally on full backbones. Since unconditional generation is not our focus, we leave the unconditional performance to App. F where we see the performance is slightly worse than RFdiffusion – as we will see, the motif-scaffolding performance is better. Both models are trained using the filtered PDB monomer dataset introduced in FrameDiff. We use the ADAM optimizer (Kingma & Ba, 2014) with learning rate 0.0001. We train each model for 6 days on 2 A6000 NVIDIA GPUs with dynamic batch sizes depending on the length of the proteins in each batch — a technique from FrameDiff.

**Sampling.** We use the Euler-Maruyama integrator with 500 timesteps for all sampling. Following the motif-scaffolding benchmark proposed in RFdiffusion, we sample 100 scaffolds for each of the 24 monomer motifs[6]. For each motif, the method must sample novel scaffolds with different lengths and different motif locations along the sequence. The benchmark measures how well a method can generalize beyond the native scaffolds for a set of biologically important motifs.

**Hyperparameters.** Our hyperparameters for neural network architecture, optimizer, and sampling steps all follow the best settings found in FrameFlow (Yim et al., 2023a). We leave hyperparameter search as a future work since it is not the focus of this work.

## 5.2 Motif-scaffolding results

**Baselines.** We consider RFdiffusion and the Twisted Diffusion Sampler (TDS) as baselines. RFdiffusion's performance is reported based on their published samples. TDS reported motif-scaffolding results with arbitrary scaffold lengths that deviated the benchmark. Therefore, we re-ran TDS with their best settings using $k = 8$ particles on the RFdiffusion benchmark. We refer to **FrameFlow-amortization** as our results with motif amortization while **FrameFlow-guidance** uses motif guidance.

**Metrics.** Previously, motif-scaffolding was only evaluated through samples passing *designability* (**Des.**). For a description of designability see App. E. Within the set of designable scaffolds, we also calculate the

---

[6]The benchmark has 25 motifs, but the motif 6VW1 involves multiple chains that FrameFlow cannot handle.

**Motif: 1QYS**     **Motif: 1YCR**     **Motif: 5TPN**

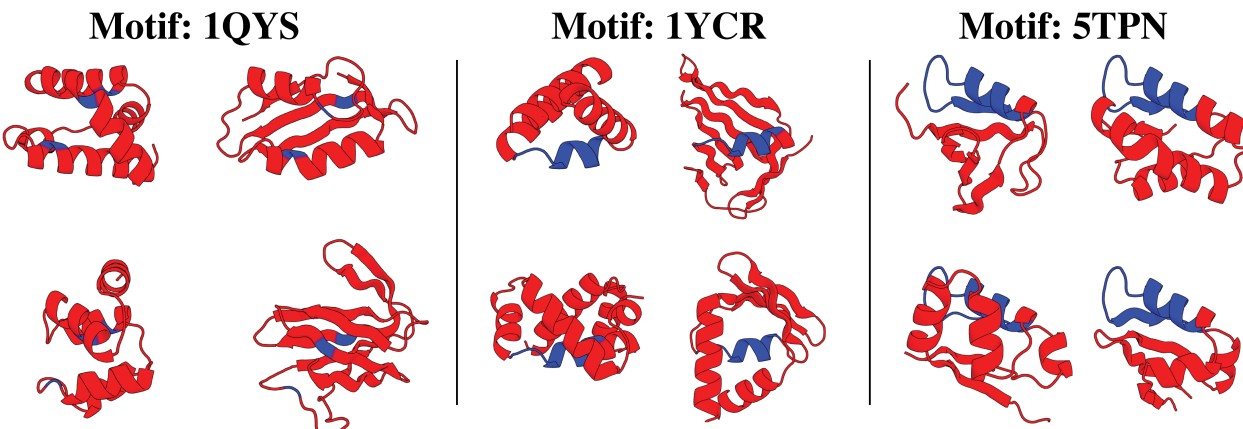

Figure 4: **FrameFlow-amortization diversity.** In blue is the motif while red is the scaffold. For each motif (1QJG, 1YCR, 5TPN), we show FrameFlow-amortization can generate scaffolds of different lengths and various secondary structure elements for the same motif. Each scaffold is in a unique cluster to showcase the samples'structural diversity.

*diversity* (**Div.**) as the number of structurally unique clusters. This is crucial since designability can be manipulated to have a 100% success rate by always sampling the same scaffold with trivial changes. In real world scenarios, diversity is desired to gain the most informative feedback from expensive wet-lab experiments (Yang et al., 2019). Thus diversity provides an additional data point to check for mode collapse where the model is sampling same scaffold repeatedly. Clusters are computed using MaxCluster (Herbert & Sternberg, 2008) with TM-score threshold set to 0.5.

**Benchmark.** Fig. 3 shows how each method fares against each other in designability and diversity on each motif of the motif-scaffolding benchmark. While it appears RFdiffusion gets lots of successful scaffolds, the number of *unique* scaffolds is far lower than both our FrameFlow approaches. TDS achieves lower designable scaffolds on average, but demonstrates strong performance on a small subset of motifs. There are some motifs that only RFdiffusion can solve (7MRX_85, 7MRX_128) while FrameFlow is able to solve cases RFdiffusion cannot (1QJG, 4JHW, 5YUI).

Table 1: Motif-scaffolding aggregate metrics

| Method | Solved (↑) | Div. (↑) | Speed (↓) |
|---|---|---|---|
| FrameFlow-amort. | **20** | **353** | 18s |
| FrameFlow-guid. | **20** | 192 | **18s** |
| RFdiffusion | **20** | 141 | 50s |
| TDS | 19 | 217 | 117s |

Tab. 1 provides the number of motifs each method solves – which means at least one designable scaffold is sampled – and the number of total designable clusters sampled across all motifs. Here we see each method can solve 19-20 solves motifs, but FrameFlow-amortization can achieve nearly double the number of unique scaffolds (clusters) as RFdiffusion. FrameFlow-amortization outperforms FrameFlow-guidance on diversity. A potential reason for the improved diversity is the use of SE(3) flow matching in the unconditional model whereas TDS uses SE(3) diffusion (Yim et al., 2023b). Bose et al. (2023) found SE(3) flow matching to provide far better designability and diversity than its diffusion counterpart. Empirically, it is known flow matching outperforms diffusion on Riemannian manifolds (Chen & Lipman, 2023).

In the last column we give the number of seconds to sample a length 100 protein on a A6000 Nvidia GPU with each method. Both FrameFlow methods are significantly faster than RFdiffusion and TDS. TDS is notably slower since its run time scales with its number of particles. We conclude that FrameFlow-amortization matches RFdiffusion and TDS on the number of solved motifs while achieving much higher diversity and faster inference.

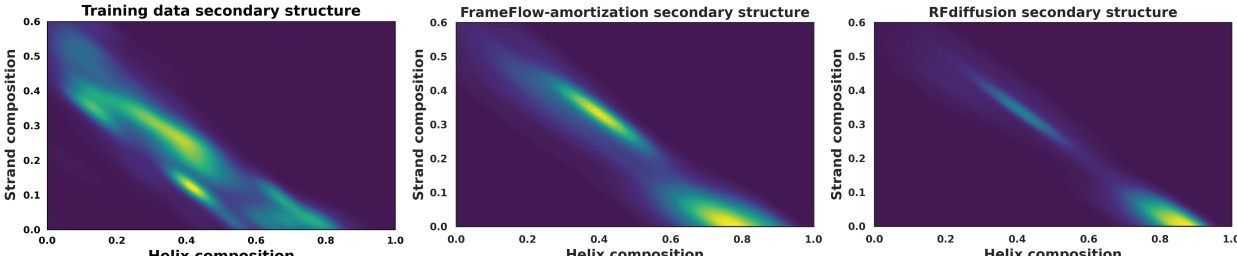

Figure 5: **Secondary structure analysis.** 2D kernel density plots of secondary structure composition of *designable* motif-scaffolds from FrameFlow-amortization and RFdiffusion. Here we see RFdiffusion tends to mostly generate helical scaffolds while FrameFlow-amortization gets much more scaffolds with strands.

**Diversity analysis.**   To visualize the diversity of the scaffolds, Fig. 4 shows several of the clusters for motifs 1QJG, 1YCR, and 5TPN where FrameFlow can generate significantly more clusters than RFdiffusion. Each scaffold demonstrates a wide range of secondary structure elements across multiple lengths. To quantify this in more depth, Fig. 5 plot the helical and strand compositions (computed with DSSP (Kabsch & Sander, 1983)) of designable motif-scaffolds from FrameFlow-amortization compared to RFdiffusion. We see FrameFlow-amotization achieves a better spread of secondary structure components than RFdiffusion. A potential reason for RFdiffusion's overall lower diversity is due to its lack of secondary structure diversity – favoring to sample mostly helical structures. App. G provides additional analysis into the FrameFlow motif-scaffolding results. We conclude FrameFlow-amortization achieves much more structural diversity than RFdiffusion.

## 6   Discussion

In this work, we present two methods building on FrameFlow for tackling motif-scaffolding. These methods can be used with any flow-based model. First, with motif-amortization we adapt the training of FrameFlow to additionally be conditioned on the motif — in effect turning FrameFlow into a conditional generative model. Second, with motif guidance, we use an unconditionally trained FrameFlow for the task of motif-scaffolding though without any additional task-specific training. We empirically evaluated both approaches, FrameFlow-amortization and FrameFlow-guidance, on the motif-scaffolding benchmark from RFdiffusion where we find both methods achieve competitive results with state-of-the-art methods. Moreover, they are able to sample more unique scaffolds and achieve higher diversity. It is important to note amortization and guidance are complementary techniques. Amortization outperforms guidance but requires conditional training while guidance can use unconditional flow models without further training. Guidance generally performs worse due to approximation error in Eq. (12) from using an unconditional model in conditional task. We stress the need to report both success rate and diversity to detect when a model suffers from mode collapse. Lastly, we caveat that all our results and metrics are computational, which may not necessarily transfer to wet-lab success.

**Future directions.**   We have extended FrameFlow for motif-scaffolding; further extensions include binder, enzyme, and symmetric design — all which RFdiffusion can currently achieve. For these capabilities, we require extending FrameFlow to handle multimeric proteins. While motif guidance does not outperform motif amortization, it is possible extending TDS to flow matching could close that gap. Related to guidance, one could explore conditioning mechanisms to control properties of the scaffold such as its secondary structure. We make use of a heuristic for Riemannian reconstruction guidance that may be further improved. Despite our progress, there still remains areas of improvement to achieve success in all 25 motifs in the benchmark.

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

# Appendix

## A  Organisation of appendices

The appendix is organized as follows. App. B provides details and derivations for FrameFlow (Yim et al., 2023a) that we introduce in Sec. 2.2. App. D provides derivation of motif guidance used in Sec. 3.2. Designability is an important metric in our experients, so we provide a description of it in App. E. Lastly, we include additional results on unconditional generation App. F and motif-scaffolding App. G.

## B  FrameFlow details

### B.1  Backbone SE(3) representation

A protein can be described by its sequence of residues, each of which takes on a discrete value from a vocabulary of amino acids, as well as the 3D structure based on the positions of atoms within each residue. The 3D structure in each residue can be separated into the backbone and side-chain atoms with the composition of backbone atoms being constant across all residues while the side-chain atoms vary depending on the amino acid assignment. For this reason, FrameFlow and previous SE(3) diffusion models (Watson et al., 2023; Yim et al., 2023b) only model the backbone atoms with the amino acids assumed to be unknown. A second model is typically used to design the amino acids after the backbone is generated. Each residue's backbone atoms follows a repeated arrangement with limited degrees of freedom due to the rigidity of the covalent bonds. AlphaFold2 (AF2) (Jumper et al., 2021) proposed a SE(3) parameterization of the backbone atoms that we show in Fig. 6. AF2 uses a mapping of four backbone atoms to a single translation and rotation that reduces the degrees of freedom in the modeling. It is this SE(3) representation we use when modeling protein backbones. We refer to Appendix I of Yim et al. (2023b) for algorithmic details of mapping between elements of SE(3) and backbone atoms.

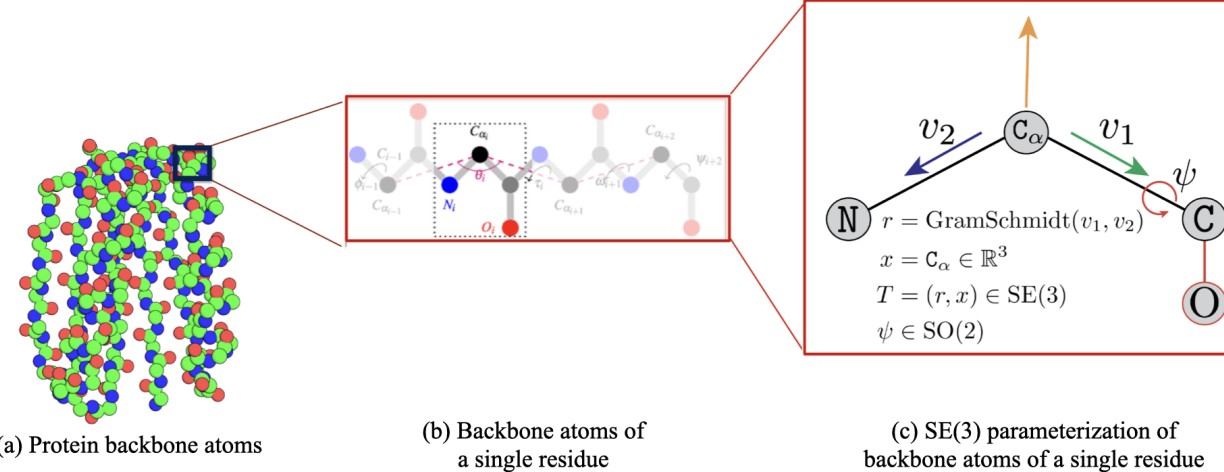

(a) Protein backbone atoms     (b) Backbone atoms of a single residue     (c) SE(3) parameterization of backbone atoms of a single residue

Figure 6: Backbone parameterization with SE(3). (a) Shows the full protein backbone atomic structure without side-chains. (b) Zooms in the backbone atoms of a single residue. Note the repeated arrangements of backbone atoms in each residue. (c) The transformation of turning each set of four backbone atoms into an element of SE(3).

### B.2  SE(3) flow matching implementation

This section provides implementation details for SE(3) flow matching and FrameFlow. As stated in Sec. 2.1, $\text{SE}(3)^N$ can be characterized as the product manifold $\text{SE}(3)^N = \mathbb{R}^N \times \text{SO}(3)^N$. It follows that flow matching on $\text{SE}(3)^N$ is equivalent to flow matching on $\mathbb{R}^{3N}$ and $\text{SO}(3)^N$. We will parameterize backbones with $N$

residues $\mathbf{T} = (\mathbf{x}, \mathbf{r}) \in \mathrm{SE}(3)^N$ by translations $\mathbf{x} \in \mathbb{R}^{3N}$ and rotations $\mathbf{r} \in \mathrm{SO}(3)^N$. As a reminder, we use bold face for vectors of all the residue: $\mathbf{T} = [T^{(1)}, \ldots, T^{(N)}]$, $\mathbf{x} = [x^{(1)}, \ldots, x^{(N)}]$, $\mathbf{r} = [r^{(1)}, \ldots, r^{(N)}]$.

Riemannian flow matching (Sec. 2.1) proceeds by defining the conditional flows,

$$x_t^{(n)} = (1-t)x_0^{(n)} + tx_1^{(n)}, \qquad r_t^{(n)} = \exp_{r_0^{(n)}}\left(t\log_{r_0^{(n)}}(r_1^{(n)})\right), \tag{16}$$

for each residue $n \in \{1, \ldots, N\}$. As priors we use $x_0^{(n)} \sim \overline{\mathcal{N}}(0, I_3)$ and $r_0^{(n)} \sim \mathcal{U}(\mathrm{SO}(3))$. $\overline{\mathcal{N}}(0, I_3)$ is the isotropic Gaussian in 3D with centering where each sample is centered to have zero mean – this is important for equivariance later on. $\mathcal{U}(\mathrm{SO}(3))$ is the uniform distribution over $\mathrm{SO}(3)$. The end points $x_1^{(n)}$ and $r_1^{(n)}$ are samples from the data distribution $p_1$.

Eq. (16) uses the geodesic path with linear interpolation; however, alternative conditional flows can be used (Chen & Lipman, 2023). A special property of $\mathrm{SO}(3)$ is that $\exp_{r_0}$ and $\log_{r_0}$ can be computed in closed form using the well known Rodrigues' formula. The corresponding conditional vector fields are

$$u_{\mathbb{R}}^{(n)}(x_t^{(n)}, t|x_1^{(n)}) = \frac{x_1^{(n)} - x_t^{(n)}}{1-t}, \qquad u_{\mathrm{SO}(3)}^{(n)}(r_t^{(n)}, t|r_1^{(n)}) = \frac{\log_{r_t^{(n)}}(r_1^{(n)})}{1-t}. \tag{17}$$

We train neural networks to regress the conditional vector fields through the following parameterization,

$$\hat{v}_{\mathbb{R}}^{(n)}(\mathbf{T}_t, t) = \frac{\hat{x}_1^{(n)}(\mathbf{T}_t) - x_t^{(n)}}{1-t}, \quad \hat{v}_{\mathrm{SO}(3)}^{(n)}(\mathbf{T}_t, t) = \frac{\log_{r_t^{(n)}}(\hat{r}_1^{(n)}(\mathbf{T}_t))}{1-t}, \tag{18}$$

where the neural network outputs the *denoised* predictions $\hat{x}_1^{(n)}$ and $\hat{r}_1^{(n)}$ while using the noised backbone $\mathbf{T}_t$ as input. We now modify the loss from Eq. (5) with practical details from FrameFlow,

$$\mathcal{L}_{\mathrm{SE}(3)} = \mathbb{E}_{\mathcal{U}(t;0,1),p_1(\mathbf{T}_1),p_0(\mathbf{T}_0)}\left[\mathcal{L}_{\mathbb{R}}(\mathbf{T}_t, \mathbf{T}_1, t) + 2\mathcal{L}_{\mathrm{SO}(3)}(\mathbf{T}_t, \mathbf{T}_1, t) + \mathbb{1}(t > 0.5)\mathcal{L}_{\mathrm{aux}}(\mathbf{T}_t, \mathbf{T}_1, t)\right] \tag{19}$$

$$\mathcal{L}_{\mathbb{R}}(\mathbf{T}_t, \mathbf{T}_1, t) = \|\mathbf{u}_{\mathbb{R}}(\mathbf{x}_t|\mathbf{x}_1, t) - \hat{\mathbf{v}}_{\mathbb{R}}(\mathbf{T}_t, t)\|_{\mathbb{R}}^2 = \frac{\|\mathbf{x}_1 - \hat{\mathbf{x}}_1\|_{\mathbb{R}}^2}{(1 - \min(t, 0.9))^2} \tag{20}$$

$$\mathcal{L}_{\mathrm{SO}(3)}(\mathbf{T}_t, \mathbf{T}_1, t) = \|\mathbf{u}_{\mathrm{SO}(3)}(\mathbf{r}_t|\mathbf{r}_1, t) - \hat{\mathbf{v}}_{\mathrm{SO}(3)}(\mathbf{T}_t, t)\|_{\mathrm{SO}(3)}^2 = \frac{\left\|\log_{\mathbf{r}_t^{(n)}}(\mathbf{r}_1^{(n)}) - \log_{\mathbf{r}_t^{(n)}}(\hat{\mathbf{r}}_1^{(n)}(\mathbf{T}_t))\right\|_{\mathrm{SO}(3)}^2}{(1 - \min(t, 0.9))^2}. \tag{21}$$

We up weight the $\mathrm{SO}(3)$ loss $\mathcal{L}_{\mathrm{SO}(3)}$ such that it is on a similar scale as the translation loss $\mathcal{L}_{\mathbb{R}}$. Eq. (20) is simplified to be a loss directly on the denoised predictions. Both Eq. (20) and Eq. (21) have modified denominators $(1 - \min(t, 0.9))^{-2}$ instead of $(1-t)^{-1}$ to avoid the loss blowing up near $t \approx 1$. In practice, we sample $t$ uniformly from $\mathcal{U}[\epsilon, 1]$ for small $\epsilon$. Lastly, $\mathcal{L}_{\mathrm{aux}}$ is taken from section 4.2 in Yim et al. (2023b) where they apply a RMSD loss over the full backbone atom positions and pairwise distances. We found using $\mathcal{L}_{\mathrm{aux}}$ for all $t > 0.5$ to be helpful. The remainder of this section goes over additional details in FrameFlow.

**Alternative SO(3) prior.** Yim et al. (2023a) reported using the IGSO3($\sigma = 1.5$) prior (Nikolayev & Savyolov, 1970) for $\mathrm{SO}(3)$ instead of $\mathcal{U}(\mathrm{SO}(3))$ lead to improved performance. The choice of $\sigma = 1.5$ will shift the $r_0$ samples away from $\pi$ where near degenerate solutions can arise in the geodesic. We follow using IGSO3($\sigma = 1.5$) for training while using the $\mathcal{U}(\mathrm{SO}(3))$ prior for sampling.

**Pre-alignment.** Following (Klein et al., 2023) and Shaul et al. (2023), we pre-align samples from the prior and the data by using the Kabsch algorithm to align the noise with the data to remove any global rotation that results in a increased kinetic energy of the ODE. Specifically, for translation noise $\mathbf{x}_0 \sim \mathcal{N}(0, I_3)^N$ and data $\mathbf{x}_1 \sim p_1$ where $\mathbf{x}_0, \mathbf{x}_1 \in \mathbb{R}^{3 \times N}$ we solve $r^* = \arg\min_{r \in \mathrm{SO}(3)} \|r\mathbf{x}_0 - \mathbf{x}_1\|_{\mathbb{R}}^2$ and use the *aligned* noise $r^*\mathbf{x}_0$ during training. Yim et al. (2023a) found this to aid in training efficiency which we adopt.

**Symmetries.** We perform all modelling within the zero center of mass (CoM) subspace of $\mathbb{R}^{N \times 3}$ as in Yim et al. (2023b). This entails simply subtracting the CoM from the prior sample $\mathbf{x}_0$ and all datapoints $\mathbf{x}_1$. As $\mathbf{x}_t$ is a linear interpolation between the noise sample and data, $\mathbf{x}_t$ will have 0 CoM also. This guarantees that the distribution of sampled frames that the model generates is SE(3)-invariant. To see this, note that the prior distribution is SE(3)-invariant and the learned vector field $\mathbf{v}_{\mathrm{SE}(3)}$ is equivariant because we use an SE(3)-equivariant architecture. Hence by Köhler et al. (2020), the push-forward of the prior under the flow is invariant.

**Auxiliary losses.** We use the same auxiliary losses in (Yim et al., 2023a).

**SO(3) inference scheduler.** The conditional flow in Eq. (16) uses a constant linear interpolation along the geodesic path where the distance of the current point $x$ to the endpoint $x_1$ is given by a pre-metric $d_g : \mathcal{M} \times \mathcal{M} \to \mathbb{R}$ induced by the Riemannian metric $g$ on the manifold. To see this, we first recall the general form of the conditional vector field with $x, x_1 \in \mathcal{M}$ is given as follows (Chen & Lipman, 2023),

$$u_t(x|x_1) = \frac{\mathrm{d}\log\kappa(t)}{\mathrm{d}t}d(x,x_1)\frac{\nabla d(x,x_1)}{\|\nabla d(x,x_1)\|^2} \tag{22}$$

$$= \frac{\mathrm{d}\log\kappa(t)}{\mathrm{d}t}\frac{\nabla d(x,x_1)^2}{2\|\nabla d(x,x_1)\|^2} \tag{23}$$

$$= \frac{\mathrm{d}\log\kappa(t)}{\mathrm{d}t}\frac{-\log_x(x_1)}{\|\nabla d(x,x_1)\|^2} \tag{24}$$

$$= \frac{-\mathrm{d}\log\kappa(t)}{\mathrm{d}t}\log_x(x_1), \tag{25}$$

with $\kappa(t)$ a monotonically decreasing differentiable function satisfying $\kappa(0) = 1$ and $\kappa(1) = 0$, referred as the *interpolation rate* [7]. Then plugging in a the linear schedule $\kappa(t) = 1 - t$, we recover Eq. (17)

$$u_t(x|x_1) = \frac{-\mathrm{d}\log\kappa(t)}{\mathrm{d}t}\log_x(x_1) = \frac{1}{1-t}\log_x(x_1). \tag{26}$$

However, we found this interpolation rate to perform poorly for SO(3) for inference time. Instead, we utilize an exponential scheduler $\kappa(t) = e^{-ct}$ for some constant $c$. The intuition being that for high $c$, the rotations accelerate towards the data faster than the translations which evolve according to the linear schedule. The SO(3) conditional flow in Eq. (16) and vector field in Eq. (17) become the following with the exponential schedule,

$$r_t = \exp_{r_0}\left(\left(1 - e^{-ct}\right)\log_{r_0}(r_1)\right) \tag{27}$$

$$v_r^{(n)} = c\log_{r_t^{(n)}}\left(\hat{r}_1^{(n)}\right). \tag{28}$$

We find $c = 10$ or $5$ to work well and use $c = 10$ in our experiments. Interestingly, we found the best performance when $\kappa(t) = 1 - t$ was used for SO(3) during training while $\kappa(t) = e^{-ct}$ is used during inference. We found using $\kappa(t) = e^{-ct}$ during training made training too easy with little learning happening.

The vector field in Eq. (28) matches the vector field in FoldFlow when inference *annealing* is performed (Bose et al., 2023). However, their choice of scaling was attributed to normalizing the predicted vector field rather than the schedule. Indeed they proposed to linearly scale up the learnt vector field via $\lambda(t) = (1 - t)c$ at sampling time, i.e. to simulate the following ODE:

$$\mathrm{d}r_t = \lambda(t)v(r_t, t)\mathrm{d}t.$$

However, as hinted at earlier, this is equivalent to using at sampling time a different vector field $\tilde{v}(r_t, t)$—induced by an *exponential* schedule $\tilde{\kappa}(t) = e^{-ct}$—instead of the *linear* schedule $\kappa(t) = 1 - t$ (that the neural

---

[7]$\kappa(t)$ acts as a scheduler that determines the rate at which $d(\cdot|x_1)$ decreases, since we have that $\phi_t$ decreases $d(\cdot, x_1)$ according to $d(\phi_t(x_0|x_1), x_1) = \kappa(t)d(x_0, x_1)$(Chen & Lipman, 2023).

network $\hat{r}_1^\theta$ was trained with). Indeed we have

$$\tilde{v}(r_t, t) = -\partial_t \log \tilde{\kappa}(t) \log_{r_t}(\hat{r}_1) = -\frac{-\partial_t \log \tilde{\kappa}(t)}{-\partial_t \log \kappa(t)} \partial_t \log \kappa(t) \log_{r_t}(\hat{r}_1) \tag{29}$$

$$= -c(1-t)\partial_t \log \kappa(t) \log_{r_t}(\hat{r}_1) = c(1-t) \, v(r_t, t) = \lambda(t) \, v(r_t, t). \tag{30}$$

## C  Data augmentation

---
**Algorithm 1** Motif-scaffolding data augmentation

---
**Require:** Protein backbone $\mathbf{T}$; Min and max motif percent $\gamma_{\min} = 0.05$, $\gamma_{\max} = 0.5$.

1: $s \sim \text{Uniform}\{\lfloor N \cdot \gamma_{\min} \rfloor, \ldots, \lfloor N \cdot \gamma_{\max} \rfloor\}$        ▷ Sample maximum motif size.
2: $m \sim \text{Uniform}\{1, \ldots, s\}$        ▷ Sample maximum number of motifs.
3: $\mathbf{T}^M \leftarrow \emptyset$
4: **for** $i \in \{1, \ldots, m\}$ **do**
5:      $j \sim \text{Uniform}\{1, \ldots, N\} \setminus \mathbf{T}^M$        ▷ Sample location for each motif
6:      $\ell \sim \text{Uniform}\{1, \ldots, s - m + i - |\mathbf{T}^M|\}$        ▷ Sample length of each motif.
7:      $\mathbf{T}^M \leftarrow \mathbf{T}^M \cup \{T_j, \ldots, T_{\min(j+\ell, N)}\}$        ▷ Append to existing motif.
8: **end for**
9: $\mathbf{T}^S \leftarrow \{T_1, \ldots, T_N\} \setminus \mathbf{T}^M$        ▷ Assign rest of residues as the scaffold
10: **return** $\mathbf{T}^M, \mathbf{T}^S$

---

## D  Motif guidance details

For the sake of completeness, we derive in this section the guidance term in Eq. (8) for the flow matching setting. In particular, we want to derive the conditional vector field $v(x_t, t|y)$ in terms of the unconditional vector field $v(x_t, t)$ and the correction term $\nabla \log p_t(y|x_t)$. Beware, in the following we adopt the time notation from diffusion models, i.e. $t = 0$ for denoised data to $t = 1$ for fully noised data. We therefore need to swap $t \rightarrow 1 - t$ in the end results to revert to the flow matching notations.

Let's consider the process associated with the following noising stochastic differential equation (SDE)

$$\mathrm{d}x_t = f(x_t, t)\mathrm{d}t + g(t)\mathrm{d}B_t \tag{31}$$

which admits the following time-reversal denoising process

$$\mathrm{d}x_t = \left[ f(x_t, t) - g(t)^2 \nabla \log p_t(x_t) \right] \mathrm{d}t + g(t)\mathrm{d}B_t. \tag{32}$$

Thanks to the Fokker-Planck equation, we know that the the following ordinary differential equation admits the same marginal as the SDE Eq. (32):

$$\mathrm{d}x_t = \left[ f(x_t, t) - \frac{1}{2}g(t)^2 \nabla \log p_t(x_t) \right] \mathrm{d}t$$

$$= v(x_t, t)\mathrm{d}t. \tag{33}$$

with $v(x_t, t)$ being the probability flow vector field.

Now, conditioning on some observation $y$, we have

$$\mathrm{d}x_t = v(x_t, t|y)\mathrm{d}t$$

$$= \left[ f(x_t, t) - \frac{1}{2}g(t)^2 \nabla \log p_t(x_t|y) \right] \mathrm{d}t$$

$$= \left[ f(x_t, t) - \frac{1}{2}g(t)^2 \left( \nabla \log p_t(x_t) + \nabla \log p_t(y|x_t) \right) \right] \mathrm{d}t$$

$$= \left[ v(x_t, t) - \frac{1}{2}g(t)^2 \nabla \log p_t(y|x_t) \right] \mathrm{d}t. \tag{34}$$

Eq. (34) follows from the same reverse SDE theory of Eq. (32) except the initial state distribution is $p(x_t|y)$). The drift $f(x_t, t)$ and diffusion $g(t)$ coefficients are unchanged while only the score reflect the new initial distribution. More details can be found in App. I of Song et al. (2020). We only need to know $g(t)$ to adapt reconstruction guidance–which estimates $\nabla \log p_t(y|x_t)$–to the flow matching setting where we want to correct the vector field. Given a particular choice of interpolation $x_t$ from flow matching, let's derive the associated $g(t)$.

**Euclidean setting**  Assume $x_0$ is data and $x_1$ is noise, with $x_1 \sim \mathcal{N}(0, \mathrm{I})$. In Euclidean flow matching, we assume a linear interpolation $x_t = (1 - t)x_0 + tx_1$. Conditioning on $x_0$, we have the following conditional marginal density $p_{t|0} = \mathcal{N}\left((1 - t)x_0, t^2\mathrm{I}\right)$. Meanwhile, let's derive the marginal density $\tilde{p}_{t|0}$ induced by Eq. (31). Assuming a linear drift $f(x_t, t) = \mu(t)x_t$, we know that $\tilde{p}_{t|0}$ is Gaussian. Let's derive its mean $m_t = \mathbb{E}[x_t]$ and covariance $\Sigma_t = \mathrm{Cov}[x_t]$. We have that (Särkkä & Solin, 2019)

$$\frac{\mathrm{d}}{\mathrm{d}t}m_t = \mathbb{E}\left[f(x_t, t)\right] = \mu(t)m_t. \tag{35}$$

thus

$$\mathbb{E}[x_t] = \exp\left(\int_0^t \mu(s)\mathrm{d}s\right) x_0. \tag{36}$$

Additionally,

$$\frac{\mathrm{d}}{\mathrm{d}t}\Sigma_t = \mathbb{E}\left[f(x_t, t)(m_t - x_t)^\top\right] + \mathbb{E}\left[f(x_t, t)^\top(m_t - x_t)\right] + g(t)^2\mathrm{I} \tag{37}$$

$$= 2\mu(t)\Sigma_t + g(t)^2\mathrm{I}, \tag{38}$$

Matching $\tilde{p}_{t|0}$ and $p_{t|0}$, we get

$$\exp\left(\int_0^t \mu(s)\mathrm{d}s\right) x_0 = (1 - t)x_0 \tag{39}$$

$$\Leftrightarrow \int_0^t \mu(s)\mathrm{d}s = \ln(1 - t) \tag{40}$$

$$\Leftrightarrow \mu(t) = -\frac{1}{1 - t} \tag{41}$$

and

$$2\mu(t)t^2 + g(t)^2 = 2t \tag{42}$$

$$\Leftrightarrow -2\frac{1}{1 - t}t^2 + g(t)^2 = 2t \tag{43}$$

$$\Leftrightarrow g(t)^2 = 2t + 2\frac{1}{1 - t}t^2 \tag{44}$$

$$\Leftrightarrow g(t)^2 = \frac{2t}{1 - t}. \tag{45}$$

The equivalent SDE that gives the same marginals is Therefore, the following SDE gives the same conditional marginal as flow matching:

$$\mathrm{d}x_t = \frac{-1}{1 - t}x_t\mathrm{d}t + \sqrt{\frac{2t}{1 - t}}\mathrm{d}\mathrm{B}_t. \tag{46}$$

**SO(3) setting**  The conditional marginal density $\tilde{p}_{t|0}$ induced by Eq. (31) with zero drift $f(r_t, t) = 0$ is given by the IGSO(3) distribution (Yim et al., 2023b): $\tilde{p}_{t|0} = \mathrm{IGSO}(3)(r_t, r_0, t)$. We are not aware of a closed form formula for the variance of such a distribution.

On the flow matching side, we assume $r_0$ is data and $r_1$ is noise, with $r_1 \sim \mathcal{U}(\mathrm{SO}(3))$, and a geodesic interpolation $r_t = \exp_{r_0}(t \log_{r_0}(r_1))$. We posit that the induced conditional marginal $p_{t|0}$ is *not* an IGSO(3) distribution. As such, it appears non-trivial to derive the required equivalent diffusion coefficient $g(t)$ for SO(3). We therefore use as a heuristic the same $g(t)$ as for $\mathbb{R}^d$.

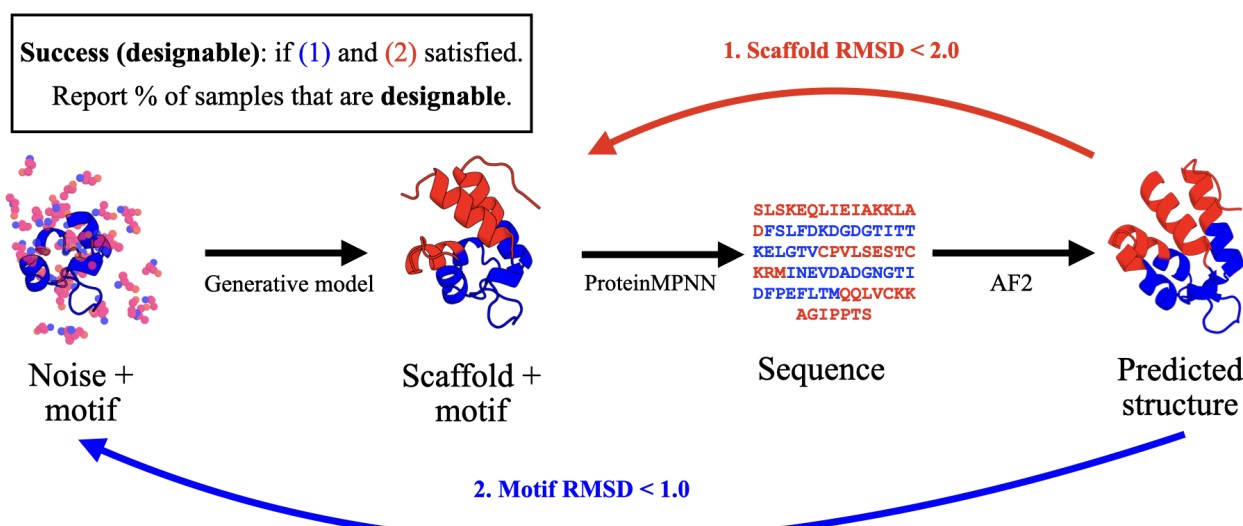

Figure 7: Schematic of computing motif-scaffolding designability. "Generative model" is a stand-in for the method used to generate scaffolds conditioned on the motif. From there, we use ProteinMPNN (Dauparas et al., 2022) to design the sequence then use AlphaFold2 (AF2) (Jumper et al., 2021) in predicting the structure of the sequence. The RMSD is calculated on the scaffold (blue) and motif (red) separately with alignments. A generated scaffold passes designability if the scaffold RMSD < 2.0 and motif RMSD < 1.0.

## E Designability

We provide details of the designability metric for motif-scaffolding and unconditional backbone generation previously used in prior works (Watson et al., 2023; Wu et al., 2023). The quality of a backbone structure is nuanced and difficult to find a single metric for. One approach that has been proven reliable in protein design is using a highly accurate protein structure prediction network to recapitulate the structure after the sequence is inferred. Prior works (Bennett et al., 2023; Wang et al., 2021) found the best method for filtering backbones to use in wet-lab experiments was the combination of ProteinMPNN (Dauparas et al., 2022) to generate the sequences and AlphaFold2 (AF2) (Jumper et al., 2021) to recapitulate the structure. We choose to use the same procedure in determining the computational success of our backbone samples which we describe next. As always, we caveat these results are computational and may not transfer to wet-lab validation. While ESMFold (Jumper et al., 2021) may be used in place of AF2, we choose to follow the setting of RFdiffusion as close as possible.

We refer to *sampled backbones* as backbones generated from our generative model. Following RFdiffusion, we use ProteinMPNN at temperature 0.1 to generate 8 sequences for each backbone in motif-scaffolding and unconditional backbone generation. In motif-scaffolding, the motif amino acids are kept fixed – Protein-MPNN only generates amino acids for the scaffold. The *predicted backbone* of each sequence is obtained with the fourth model in the five model ensemble used in AF2 with 0 recycling, no relaxation, and no multiple sequence alignment (MSA) – as in the MSA is only populated with the query sequence. Fig. 7 provides a schematic of how we compute designability for motif-scaffolding. A sampled backbone is successful or referred to as designable based on the following criterion depending on the task:

- **Unconditional backbone generation**: successful if the Root Mean Squared Deviation (RMSD) of all the backbone atoms is < 2.0Åafter global alignment of the Carbon alpha positions.

- **Motif-scaffolding**: successful if the RMSD of motif atoms is < 1Åafter alignment on the motif Carbon alpha positions. Additionally, the RMSD of the scaffold atoms must be < 2Åafter alignment on the scaffold Carbon alpha positions.

## F    FrameFlow unconditional results

We present backbone generation results of the unconditional FrameFlow model used in FrameFlow-guidance. We do not perform an in-depth analysis since this task is not the focus of our work. Characterizing the backbone generation performance ensures we are using a reliable unconditional model for motif-scaffolding. We evaluate the unconditionally trained FrameFlow model by sampling 100 samples from lengths 70, 100, 200, and 300 as done in RFdiffusion. The results are shown in Tab. 2. We find that FrameFlow achieves slightly worse designability while achieving improved novelty. We conclude that FrameFlow is able to achieve strong unconditional backbone generation results that are on par with a current state-of-the-art unconditional diffusion model RFdiffusion. We perform secondary structure analysis of the unconditional samples in Fig. 8.

Table 2: Unconditional generation metrics.

| Method | Des.(↑) | Div. (↑) | Nov. (↓) |
|---|---|---|---|
| FrameFlow | 0.86 | 155 | **0.61** |
| RFdiffusion | **0.89** | **159** | 0.65 |

We find FrameFlow has a tendency to sample more alpha helical structures than the data distribution but still has roughly the same coverage of structures. Future work could investigate the cause of such helical tendency and improve the secondary structure sample distribution.

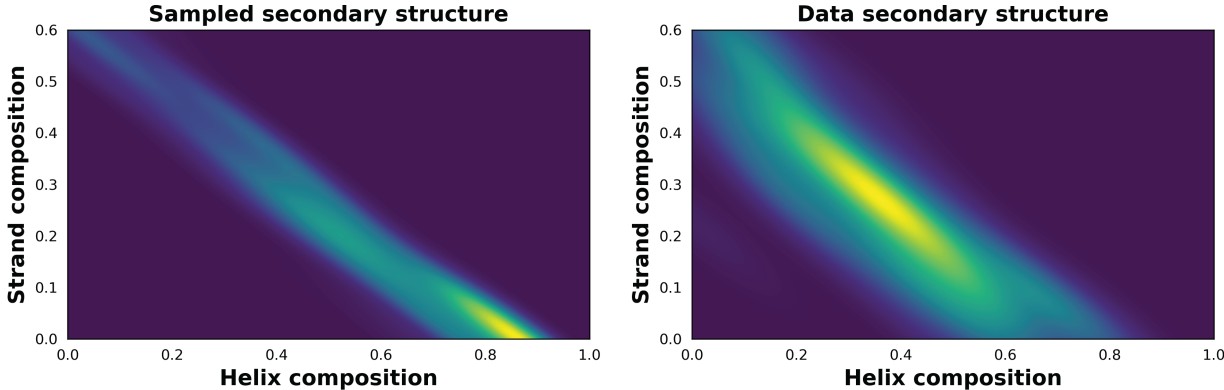

Figure 8:  Secondary structure analysis of unconditional samples. Using FrameFlow we sample 100 proteins for each length 70, 100, 200, and 300. The 2D kernel density estimation plot of the secondary structure composition is shown on left. On the right, we show the secondary structure composition of all length 70, 100, 200, and 300 proteins in the training set. We find FrameFlow has a tendency to sample more alpha helical structures than the data distribution.

## G    FrameFlow motif-scaffolding analysis

In this section, we provide additional analysis into the motif-scaffolding results in Sec. 5.2. Our focus is on analyzing the motif-scaffolding with FrameFlow: motif amortization and guidance. The first analysis is the empirical cumulative distribution functions (ECDF) of the motif and scaffold RMSD shown in Fig. 9. We find that the main advantage of amortization is in having a higher percent of samples passing the motif RMSD threshold compared to the scaffold RMSD. Amortization has better scaffold RMSD but the gap is smaller than motif RMSD. The ECDF curves are roughly the same for both methods.

Tab. 3 shows the average pairwise TM-score for all designable scaffolds per motif. We report this for each method where we find our FrameFlow approaches get the lowest average pairwise TM-score in 19 out of 24 motifs. The average pairwise TM-score is meant to complement the cluster criterion for diversity in case where clustering leads to pathological behaviors. Both metrics have their strengths and weaknesses but together help provide more details on sample diversity.

Lastly, we visualize samples from FrameFlow-amortization on each motif in the benchmark. As noted in Sec. 5.2, amortization is able to solve 21 out of 24 motifs in the benchmark. In Fig. 10, we visualize the generated scaffolds that are closest to passing designability for the 3 motifs it is unable to solve. We find the failure to be in the motif RMSD being over 1ÅRMSD. However, for 4JHW, 7MRX__60, and 7MRX__128 the motif RMSDs are 1.2, 1.1, and 1.7 respectively. This shows amortization is very close to solve all motifs in the benchmark. In Fig. 11, we show designable scaffolds for each of the 21 motifs that amortization solves. We highlight the diverse range of motifs that can be solved as well as diverse scaffolds.

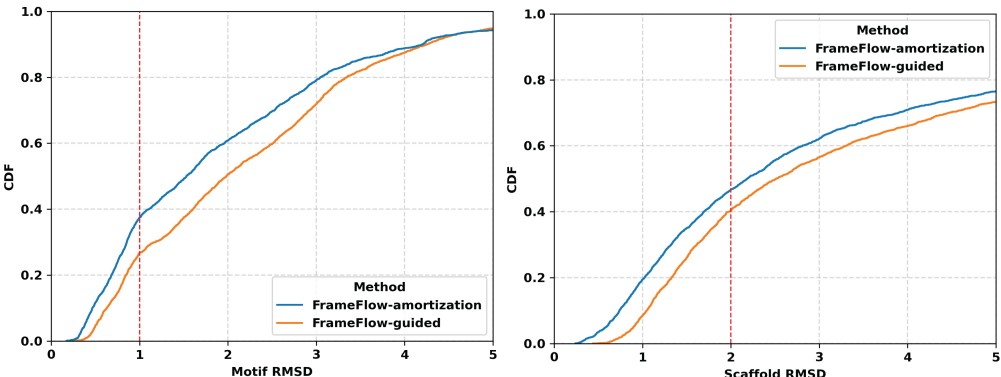

Figure 9: Empirical cumulative density plot of designability RMSD over the motif (left) and scaffold (right) for FrameFlow-amortization (blue line) and FrameFlow-guidance (orange line).

Table 3: Under each method name are average TM-score over all designable scaffolds for each motif. Lower is better which indicates more dissimilar pairwise scaffolds. N/A indicates no designable scaffolds were sampled. We find our FrameFlow approaches have the lowest average TM-scores among all methods.

| Motif | FrameFlow-amortization | FrameFlow-guidance | TDS | RFdiffusion |
|---|---|---|---|---|
| 6E6R__med | **0.3** | 0.31 | 0.36 | 0.39 |
| 2KL8 | 0.95 | **0.86** | 0.88 | 0.98 |
| 4ZYP | 0.55 | **0.44** | N/A | N/A |
| 5WN9 | 0.53 | **0.45** | 0.48 | N/A |
| 5TRV__short | 0.48 | **0.42** | 0.46 | 0.66 |
| 7MRX__60 | N/A | N/A | **0.29** | 0.59 |
| 6EXZ__short | 0.48 | 0.49 | 0.47 | **0.35** |
| 1YCR | 0.34 | **0.3** | 0.4 | 0.48 |
| 5IUS | **0.6** | N/A | N/A | 0.73 |
| 6E6R__short | 0.37 | **0.35** | 0.39 | 0.41 |
| 3IXT | **0.4** | 0.47 | 0.46 | 0.62 |
| 7MRX__85 | N/A | **0.36** | N/A | 0.56 |
| 1QJG | 0.34 | 0.44 | **0.33** | N/A |
| 1BCF | 0.76 | 0.69 | **0.5** | 0.83 |
| 5TRV__med | **0.34** | 0.37 | 0.38 | 0.43 |
| 5YUI | N/A | N/A | N/A | N/A |
| 5TPN | 0.48 | **0.54** | N/A | 0.61 |
| 1PRW | 0.75 | **0.66** | N/A | 0.76 |
| 6EXZ__med | 0.37 | 0.38 | **0.36** | 0.49 |
| 5TRV__long | **0.3** | 0.31 | N/A | 0.39 |
| 4JHW | N/A | N/A | N/A | N/A |
| 7MRX__128 | N/A | N/A | N/A | **0.5** |
| 6E6R__long | **0.28** | **0.28** | 0.46 | 0.35 |
| 6EXZ__long | **0.3** | **0.3** | 0.38 | 0.38 |

**Undesignable motif-scaffolds**

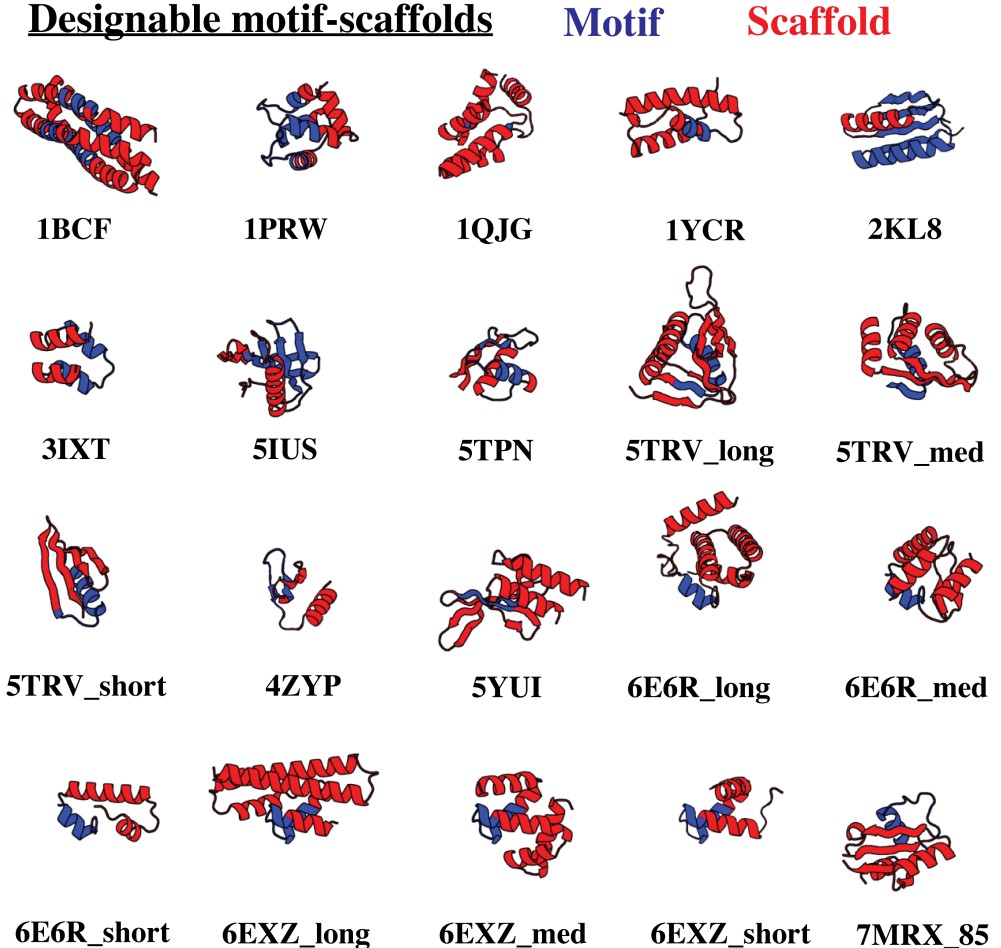

**4JHW**
Motif RMSD: 1.2
Scaffold RMSD: 1.5

**7MRX_60**
Motif RMSD: 1.1
Scaffold RMSD: 1.3

**7MRX_128**
Motif RMSD: 1.7
Scaffold RMSD: 1.4

**5WN9**
Motif RMSD: 0.75
Scaffold RMSD: 5.6

Figure 10: Closest motif-scaffolds from FrameFlow-amortization on the three motifs it fails to solve. We find the $> 1.0$Åmotif RMSD is the reason for the method failing to pass designability.

# Designable motif-scaffolds     Motif     Scaffold

**1BCF** **1PRW** **1QJG** **1YCR** **2KL8**

**3IXT** **5IUS** **5TPN** **5TRV_long** **5TRV_med**

**5TRV_short** **4ZYP** **5YUI** **6E6R_long** **6E6R_med**

**6E6R_short** **6EXZ_long** **6EXZ_med** **6EXZ_short** **7MRX_85**

Figure 11: Designable motif-scaffolds from FrameFlow-amortization on 20 out of 24 motifs in the motif-scaffolding benchmark.

