# OpenReview forum: "Improved motif-scaffolding with SE(3) flow matching"
_TMLR — Accepted by TMLR_

### Review · Reviewer_8RGg · 2024-06-16

**Summary Of Contributions:**

This paper introduces two new methods for motif-scaffolding, which is the task of building scaffolds around protein motifs. Both methods are based on FrameFlow. One, called motif amortization, starts with a pre-defined motif structure (3D positions of the alpha carbons) and generates a scaffold around it. To generate data for training, a data augmentation technique is proposed that extracts random fragments of the protein and marks them as the motif. Another method, called motif guidance, in which the motif is used to bias the generative trajectory. In the experiments, the Authors demonstrate that both methods can produce designable and diverse scaffolds, and motif amortization outperforms the other tested methods (RDdiffusion and TDS).

**Audience:**

Yes

**Broader Impact Concerns:**

I do not think any ethical implications need to be described in this paper. One potential implication would be the use of this research to create harmful biological agents.

**Claims And Evidence:**

No

**Requested Changes:**

1. Please add a short discussion about the impact of motif zero-centering on the generalizability.
2. I believe more experiments should be conducted, and better evaluation metrics should be proposed (e.g. other diversity measures as suggested above).
3. Please check carefully whether the number of clusters is not correlated with the number of designable scaffolds generated by the model.
4. The experiment should be repeated a few times to denoise the reported metrics and to provide a standard deviation.
5. An ablation study would help in understanding all the technical decisions related to the architecture and training procedure (see comments above).

**Strengths And Weaknesses:**

Strengths:
- The paper is written in a very clear way.
- The background section is helpful in understanding the study.
- The method description is supported by figures that demonstrate the diffusion process and data augmentation (although blue dots, presumably nitrogen atoms, may be confused with the parts of the blue motif)
- The theoretical and methodological part of the paper is strong, including the derivations in the appendix (although I have not checked the correctness thoroughly)

Weaknesses:
- The statement in the abstract that the proposed method achieves "a 2.5 times greater in-silico success rate of unique motif-scaffolds" is imprecise. It does not explain what "success rate" is in this context.
- The notation is inconsistent between the background and method sections: it is explicitly stated in the background section that superscripts are used to refer to residues indices, but in the next section subscripts are used.
-  I am also a little bit concerned about the zero-centering of the motif to maintain SE(3)-equivariance. Technically this seems correct, but can it impede the generalization to other motifs? For example, if you add a few atoms or another subchain to the motif, the whole structure moves which could lead to significantly different predictions.
- The parameters $\gamma_{min}$ and  $\gamma_{max}$ are not defined in Section 5.1 (only in the appendix)
- My main concern is about the experimental section. I am unsure if the chosen evaluation protocol gives the readers any meaningful information about the model performance. It seems that the proposed method generates less designable scaffolds than the already existing methods (Figure 3). The models are thus compared using scaffold diversity, but this evaluation metric is based on clustering. The number of generated designable scaffolds may impact the number of clusters. If the generated scaffold space is sparse, it would lead to more clusters as there are no scaffolds in-between that would bridge two clusters. It can be observed in Figure 3 that for the methods producing the biggest number of designable scaffolds, the number of clusters is very small (e.g. 2KL8 or 6E6R_long). It would be more convincing if the diversity metric were defined directly on the predicted atom positions, e.g. using the average RMSD between all pairs of aligned scaffolds.
- The experimental evaluation could have been done more thoroughly. It seems that the speed was measured only for one protein. There are only 24 motifs in the testing set. Maybe this evaluation should be at least repeated a few times to make sure that the noise related to random sampling is removed. No hyperparameter tuning was conducted.
- Some claims in Appendix B.2 are not supported by any experiments. For example, you say that using the IGSO3 prior improved performance, and the pre-alignment of the noise resulted in improved training efficiency. These decisions seem interesting and important for your presented methods. Maybe these experiments could be described as an ablation study.
- In appendix F, you say that FrameFlow achieves improved diversity, but Table 2 shows otherwise.

In conclusion, the approach described in the paper is very interesting, but the experiments seem rather preliminary at this stage.

---

> ### Author Response · Authors · 2024-06-29
> **Rebuttal (part 1)**
>
> We thank the reviewer for their constructive feedback and time. We are glad the reviewer acknowledged the clarity and methodological rigour of our work. We hope our response below alleviate concerns regarding the experiments.
>
> > The statement in the abstract that the proposed method achieves "a 2.5 times greater in-silico success rate of unique motif-scaffolds" is imprecise. It does not explain what "success rate" is in this context.
>
> Thanks for pointing this out! We have modified the introduction to define and discuss the concept of “in-silico success”. We have also removed the mention of success in the abstract and rewritten it as, “On a benchmark of 24 biologically meaningful motifs, we show our method achieves 2.5 times more designable and unique motif-scaffolds compared to state-of-the-art.”
>
> > The notation is inconsistent between the background and method sections: it is explicitly stated in the background section that superscripts are used to refer to residues indices, but in the next section subscripts are used.
>
> Thank you for this catch! We have now corrected the method section to use superscripts for the residue indices.
>
> > I am also a little bit concerned about the zero-centering of the motif to maintain SE(3)-equivariance. Technically this seems correct, but can it impede the generalization to other motifs? For example, if you add a few atoms or another subchain to the motif, the whole structure moves which could lead to significantly different predictions.
>
> The zero-centering of the motif serves two purposes. First, it is a requirement to maintain SE(3) equivariance of the generative model. Second, it is important to prevent the model from memorizing locations of the motif when generating new scaffolds. For instance, if we were to center on the scaffold then the model may memorize the distance between the motif and gaussian noise since the distances give away where the scaffold should go. This would hinder generalization because the model would rely on using the distance between the noise and motif to determine the scaffold. While there may be better centering strategies, we think zero centering of the motif is the most reasonable approach to start with. We have added this discussion to the end of section 3.1.0.
>
> > The parameters $\gamma_{min}$ and $\gamma_{max}$ are not defined in Section 5.1 (only in the appendix)
>
> Perhaps this was a typo? We define $\gamma_{min}=0.05$ and $\gamma_{max}=0.5$ at the very beginning of section 5.1. For convenience, we have specified their values again in appendix D.
>
> > My main concern is about the experimental section.
>
> Thank you for raising your concerns. We address them below.
>
> > I am unsure if the chosen evaluation protocol gives the readers any meaningful information about the model performance. It seems that the proposed method generates less designable scaffolds than the already existing methods (Figure 3).
>
> We rely on Figure 3 to argue that more designable scaffolds does not equate to more unique scaffolds. What ultimately matters is the proportion of unique designable scaffolds. Indeed, a method could sample the same designable scaffold repeatedly and achieve 100% success rate but it is better to sample 2 different designable scaffolds and have only 2% success. Henceforth, we argue for the number (or proportion) of unique designable scaffolds to be a much more meaningful metric. The caption has been rewritten to emphasize this point.

---

> > ### Author Response · Authors · 2024-06-29
> > **Rebuttal (part 2)**
> >
> > > The models are thus compared using scaffold diversity, but this evaluation metric is based on clustering. The number of generated designable scaffolds may impact the number of clusters. If the generated scaffold space is sparse, it would lead to more clusters as there are no scaffolds in-between that would bridge two clusters. It can be observed in Figure 3 that for the methods producing the biggest number of designable scaffolds, the number of clusters is very small (e.g. 2KL8 or 6E6R_long). It would be more convincing if the diversity metric were defined directly on the predicted atom positions, e.g. using the average RMSD between all pairs of aligned scaffolds.
> >
> > We agree that clustering has limitations as described by the reviewer and are happy to include discussion on pairwise distances between designable scaffolds as an additional metric for diversity. Clustering and pairwise distances provide complementary ways of analyzing diversity. Clusters are interpretable for structural biologists who tend to prefer using clusters when analyzing proteins. For instance, the popular AlphaFold2 database measures diversity through structural clustering [1]. Furthermore, clusters avoid giving a biased view in the case where there are, say, two far apart clusters that have high average RMSD distance but low diversity since there are only two clusters. To provide the alternative distance based view, we compute the average pairwise TM-score (which can be viewed as length normalized RMSD) between all designable scaffolds for each motif. We have also added the following text to Appendix G.
> >
> > “Tab. 3 shows the average pairwise TM-score for all designable scaffolds per motif. We report this for each
> > method where we find our FrameFlow approaches get the lowest average pairwise TM-score in 19 out of
> > 24 motifs. The average pairwise TM-score is meant to complement the cluster criterion for diversity in case clustering leads to pathological behaviors. Both metrics have their strengths and weaknesses such as clusters being interpretable  but together help provide more details on sample diversity.”
> >
> > We are unsure about this observation “in Figure 3 that for the methods producing the biggest number of designable scaffolds, the number of clusters is very small (e.g. 2KL8 or 6E6R_long).” For 2KL8 and 6E6R_long, we see that methods with both low and large number of designable scaffolds have a very small number of clusters, not just RFdiffusion. We are unable to see this trend in general across the remaining motifs. For instance, in 6EXZ_short we see the method with the most designable scaffolds has the most number of clusters.
> >
> > > The experimental evaluation could have been done more thoroughly. It seems that the speed was measured only for one protein. There are only 24 motifs in the testing set. Maybe this evaluation should be at least repeated a few times to make sure that the noise related to random sampling is removed.
> >
> > We would like to emphasize that our evaluation follows the RFdiffusion motif-scaffolding benchmark with one exception of leaving out one motif since it is a multimer target. All other details follow exactly. The 24 motifs were selected as previously designed scaffolds for biologically meaningful motifs. Each metric reported in Table 1 is estimated over the 24 motifs and with 100 scaffolds per motif. Hence, the evaluation has indeed been repeated many times over the 100 scaffolds per motif to remove noise related to random sampling.
> >
> > > No hyperparameter tuning was conducted.
> >
> > One of the benefits of our approach is that we build upon the FrameFlow model and introduce minimal changes, enabling motif-scaffolding with minimal additional hyperparameters. Our aim was not to improve FrameFlow but to do a controlled study of two motif-scaffolding techniques. We take all modeling and flow matching hyperparameters from prior works [2]. We introduce two hyperparameters: $\gamma_{max}$ and $\gamma_{min}$ which are the lower and upper bounds of the scaffold percentage during data augmentation. These are quite intuitive hyperparameters where we set $\gamma_{min}=0.05$ such that at least a few residues are always set as the motif and $\gamma_{min}=0.5$ such that at most half of the residues can be the motif otherwise the training task is too easy.
> >
> > It is possible that further hyperparameter tuning of the FrameFlow architecture or training might lead to some further yet marginal improvements. This is somewhat outside the scope of this work, so we leave this for follow up work.

---

> > > ### Author Response · Authors · 2024-06-29
> > > **Rebuttal (part 3)**
> > >
> > > > Some claims in Appendix B.2 are not supported by any experiments. For example, you say that using the IGSO3 prior improved performance, and the pre-alignment of the noise resulted in improved training efficiency. These decisions seem interesting and important for your presented methods. Maybe these experiments could be described as an ablation study.
> > >
> > > Everything in Appendix B.2 is taken from FrameFlow [2] where they found IGSO3 and pre-alignment to improve performance. These findings are also noted in [3]. We agree that these methodological choices are worthy of being further investigated, yet we leave this for future work.  Again, improving or investigating FrameFlow is not the focus of this study, this work takes FrameFlow as a given.
> > >
> > > > In appendix F, you say that FrameFlow achieves improved diversity, but Table 2 shows otherwise.
> > >
> > > Thank you for catching this! We have corrected this to only say FrameFlow achieves improved novelty.
> > >
> > > > Please add a short discussion about the impact of motif zero-centering on the generalizability.
> > >
> > > As previously discussed, we have added the following text to the end of section 3.1.0: “Zero-centering the motif also prevents the model from using the motif offset from the origin to memorize scaffold locations which helps generalization.”
> > >
> > > > I believe more experiments should be conducted, and better evaluation metrics should be proposed (e.g. other diversity measures as suggested above).
> > >
> > > Please see our discussion above. We have followed the only known motif-scaffolding benchmark taken from a seminal paper published in Nature [4]. Initially we evaluated diversity through clustering but have now included the average pairwise TMscore as well in Appendix G. We believe that the combination of clusters and TMscore provides better insight into the diversity of samples.
> > >
> > > > Please check carefully whether the number of clusters is not correlated with the number of designable scaffolds generated by the model.
> > >
> > > Based on our discussion above, we have calculated the correlation between the number of designable scaffolds and clusters across all motifs and have found there to actually be a positive correlation (Pearson 0.56 and Spearman 0.77) with p-value <0.05.
> > >
> > > > The experiment should be repeated a few times to denoise the reported metrics and to provide a standard deviation.
> > >
> > > We follow the evaluation guidelines set in the peer-reviewed RFdiffusion [4]. In the benchmark, we sample 100 scaffolds per motif to reduce the estimator variance. Combined with running ProteinMPNN and AlphaFold2 for evaluation, this is already quite computationally intense. We argue this removes the need for multiple runs due to the high number of samples per motif. One could bootstrap the 100 scaffolds into multiple runs and take standard errors but we believe this is not necessary and would like to follow RFdiffusion’s protocol.
> > >
> > > > An ablation study would help in understanding all the technical decisions related to the architecture and training procedure (see comments above).
> > >
> > > We have noted that all technical decisions related to the architecture and training procedure (other than data augmentation) are taken from prior works. These aspects were not the focus of our work and we chose to keep them constant in order to focus on investigating two complementary approaches, amortization and guidance, for motif-scaffolding.
> > >
> > > # References
> > >
> > > [1] Barrio-Hernandez, Inigo, et al. "Clustering predicted structures at the scale of the known protein universe." Nature 622.7983 (2023): 637-645.
> > >
> > > [2] Yim, Jason, et al. "Fast protein backbone generation with SE (3) flow matching." arXiv preprint arXiv:2310.05297 (2023).
> > >
> > > [3] Bose, Avishek Joey, et al. "Se (3)-stochastic flow matching for protein backbone generation." arXiv preprint arXiv:2310.02391 (2023).
> > >
> > > [4] Watson, Joseph L., et al. "De novo design of protein structure and function with RFdiffusion." Nature 620.7976 (2023): 1089-1100.

---

> > > > ### Comment · Reviewer_8RGg · 2024-07-02
> > > >
> > > > Thank you for your detailed responses and for additional experiments that show other diversity metrics. Most of my concerns were resolved. I have only two more comments:
> > > > 1. Sorry, I think my comment about the hyperparameter definition was unclear. When I said that $\gamma_{min}$ and $\gamma_{max}$ are not defined, I did not mean their values but rather a description of these hyperparameters. Based on Algorithm 1 in the Appendix, I think these parameters are minimal and maximal lengths of the motif expressed as a fraction of the sequence length, but this is not explained in the main text.
> > > > 2. I reread the FrameFlow paper and found the same statements as included in Appendix B.2. I was confused about these claims because they are written in the first-person narrative and not supported by any experiments, so I assumed these are your additions to your modified FrameFlow implementation in this paper. Please make sure these findings are not attributed to this paper by either rewriting this section or removing these statements and referring to the original paper for the details on the design decisions.

---

> > > > > ### Author Response · Authors · 2024-07-03
> > > > > **Author response**
> > > > >
> > > > > Thank you for the continued engagement! We are glad most of the reviewer concerns were addressed. Below we address the remaining concerns.
> > > > >
> > > > > > Description of augmentation hyperparameters $\gamma_{min}$ and $\gamma_{max}$
> > > > >
> > > > > Thank you for catching this missing description. We have described these hyperparameters in section 3.1.1 by adding the following: “The length of each motif is randomly sampled such that the total number of motif residues is between γmin and γmax percent of all the residues. We use $\gamma_{min}$ = 0.05 and $\gamma_{max}$ = 0.5 to ensure at least a few residues are used as the motif but not more than half the protein.”
> > > > >
> > > > > > Source of FrameFlow modeling choices
> > > > >
> > > > > We agree our choice of wording was ambiguous. We have clarified that these modeling choices are from FrameFlow in Appendix B.2 and removed first-person statements.
> > > > >
> > > > > “Alternative SO(3) prior. Yim et al. (2023a) reported using the IGSO3(σ = 1.5) prior (Nikolayev &
> > > > > Savyolov, 1970) for SO(3) instead of U(SO(3)) lead to improved performance…”
> > > > >
> > > > > “Pre-alignment… Yim et al. (2023a) found this to aid in training efficiency which we adopt.”
> > > > >
> > > > >
> > > > > We hope this has addressed the remaining concerns.

---

### Review · Reviewer_QRBg · 2024-06-19

**Summary Of Contributions:**

To design a protein, starting from a motif and leveraging its knowledge of a desired function is commonly adopted. However, such approaches tend to lack structural diversity. To alleviate this issue, the paper extends FrameFlow with motif amortization, which augments the training input motif, and motif guidance, which provides training-free conditional information. The paper conducts extensive empirical studies and justifies the effectiveness of the proposed method.

**Audience:**

Yes

**Claims And Evidence:**

Yes

**Requested Changes:**

1. Provide discussion on the approximation error and experiment with different choices of PDB.
2. Provide secondary structure analysis on FrameFlow.
3. Provide more comparison with the SOTA method, including diffusion-based and motif-scafolding based methods.
4. Provide discussion on the potential reason why the diffusion-based method suffers from low diversity scaffolding and how the flow matching based method resolves that.

**Strengths And Weaknesses:**

## Pros

- The proposed motif amortization and motif guidance are novel and effective in the protein design domain. The paper leverages SE(3) flow matching to model protein backbones, extending it to motif-scaffolding.
- The proposed methods aim to overcome the limitations of structural diversity in generated scaffolds, which is crucial for successful wet-lab validations. Extensive experiments have been conducted to provide a good insight into the components of the proposed method.
- The paper is generally well-written, with clear illustrations and tables.

## Cons

-  In Sec. 3.1, the motif amortization introduces unlabled structure from PDB as an approximation for the $p(\textbf{T}^M)$ and $p(\textbf{T}_1^S|\textbf{T}^M)$. However, the paper does not discuss the approximation error. In addition, the experiment section does not provide empirical results on the robustness of the chosen PDB.
-  The motif guidance seems to heavily depend on the TDS, a motif-guide diffusion model, as described in Sec. 3.2. The paper claims that the diffusion model based method suffers from low scaffold diversity. However, the underlying cause for such an issue, and the reason for the proposed method to alleviate it are not well discussed in the paper.
-  The writing can be improved by introducing more background knowledge on flow matching and ODE functions, as the current version might not be friendly enough for those who are not familiar with these methods. In addition, The paper only compare two motif-scafolding method in the experiment seciton, which might not convicinen enough.

---

> ### Author Response · Authors · 2024-06-29
> **Rebuttal (part 1)**
>
> We thank the reviewer for their constructive feedback and time. We are glad the reviewer acknowledged the novelty and clarity of our work. Below we address each comment line by line.
>
> > In Sec. 3.1, the motif amortization introduces unlabled structure from PDB as an approximation for the $p(\textbf{T}^M)$ and $p(\textbf{T}_1^S|\textbf{T}^M)$. However, the paper does not discuss the approximation error. In addition, the experiment section does not provide empirical results on the robustness of the chosen PDB.
>
> We mention we do not have access to the true distributions $p(\textbf{T}^M)$ and $p(\textbf{T}_1^S|\textbf{T}^M)$ since this implies knowing all functional motifs of each protein in the PDB. The true functional sites are only known for a small subset of proteins such as antibodies (CDR loops) and certain enzymes (catalytic active sites). Rather than being selective in choosing PDB examples, our approach attempts to model all possible motif-scaffold annotations in the PDB since we cannot know the motifs in advance. We are not aware of how to measure this approximation other than to test our method out on a held-out motif-scaffolding benchmark of real motifs curated in RFdiffusion which we do in our experiments. We view our experiments as empirical evidence on the robustness and realism of our data augmentation strategy that approximates the motif-scaffold distribution.
>
> > The motif guidance seems to heavily depend on the TDS, a motif-guide diffusion model, as described in Sec. 3.2. The paper claims that the diffusion model based method suffers from low scaffold diversity. However, the underlying cause for such an issue, and the reason for the proposed method to alleviate it are not well discussed in the paper.
>
> Indeed, our motif guidance approach is inspired by TDS  along with other reconstruction guidance methods where our technical contribution is translating reconstruction guidance into flow matching. But we do not “heavily depend” on TDS since we do not perform sequential monte carlo. We have several hypotheses on why diffusion models (TDS and RFdiffusion) suffer from low scaffold diversity.
>
> * Prior works have found RFdiffusion samples mostly helical structures which we corroborate in figure 5 where most of RFdiffusion’s designable scaffolds are helical. These reasons partly explain RFdiffusion’s lack of diversity.
>
> * TDS uses FrameDiff which is an underperforming generative model compared to its flow matching counterparts FoldFlow and FrameFlow, both of which are nearly identical models. Compared to FrameDiff, FoldFlow demonstrated greater performance including diversity which would also help explain FrameFlow’s improved diversity over FrameDiff. One well-known reason is the improved performance of flows over diffusion on Riemannian manifolds [1]. Other than this, we are not sure why TDS suffers from lower diversity. We mention using TDS with FrameFlow as a future direction of research.
>
> > The writing can be improved by introducing more background knowledge on flow matching and ODE functions, as the current version might not be friendly enough for those who are not familiar with these methods.
>
> Could the reviewer clarify what additional background knowledge is missing? We include background on flow matching in section 2.1 and appendix B.2 for both the Riemannian and Euclidean cases. Our background section describes the ODE as a continuous normalizing flow [2]. We include references to seminal flow matching papers such as Lipman et al 2022 which we believe is one of the best resources to understand flow matching. We looked for missing background work but are unsure where more explanation would be needed. We are happy to address specific points or spots where the writing is not clear.
>
> > In addition, The paper only compare two motif-scafolding method in the experiment seciton, which might not convicinen enough.
>
> As mentioned in our related works, there were only two other structure-based motif-scaffolding methods with publicly available code to compare our approach to. Since then, there have been concurrent works performing motif-scaffolding [3-4]. We understand the number of baseline methods is small but we hope it is understandable since motif-scaffolding is a relatively new problem in the machine learning community.
>
> > Provide discussion on the approximation error and experiment with different choices of PDB.
>
> Based on our response above, we have added the following discussion in section 3.1.1. “The lack of functional annotations in the PDB requires training over all possible motif-scaffold annotations to handle new scenarios our method may encounter in real world scenarios. In our experiments, we evaluate how this data augmentation strategy transfers to real motif-scaffolding tasks. A similar strategy is used in image infilling where image based diffusion models are trained to infill randomly masked crops of images to approximate real image infilling scenarios. (Saharia et al., 2022)“

---

> > ### Author Response · Authors · 2024-06-29
> > **Rebuttal (part 2)**
> >
> > > Provide secondary structure analysis on FrameFlow.
> >
> > We believe the reviewer is referring to analyzing the secondary structure composition of the unconditional generation performance of FrameFlow. We have added this analysis in Appendix F where we find in figure 8 that FrameFlow samples have the same secondary structure coverage as the data but with a tendency to sample more helical structures than the data. As mentioned, the bias towards helical structures is a common issue with current protein generative models [3]. We note this observation and leave it as future work to enable controllability of secondary structure composition.
> >
> > > Provide more comparison with the SOTA method, including diffusion-based and motif-scafolding based methods.
> >
> > We compare our method to RFdiffusion which is known as the state-of-the-art method for motif-scaffolding. The only other diffusion and structure based motif-scaffolding method is TDS which outperforms RFdiffusion on certain motifs. [5] is another deep learning based motif-scaffolding model but it is outperformed by RFdiffusion on all metrics. Lastly, EvoDiff [6] is a sequence based motif-scaffold model which we exclude since it utilizes multiple sequence alignments and its performance falls short of RFdiffusion on all motifs. We describe all these related works in section 4.
> >
> > As mentioned, concurrently a few recent works (post submission) have been pre-printed performing motif-scaffolding [3-4]. We are glad more works are focusing on motif-scaffolding such that more baselines can be used to compare different approaches.
> >
> > > Provide discussion on the potential reason why the diffusion-based method suffers from low diversity scaffolding and how the flow matching based method resolves that.
> >
> > Based on our response above, we have added the following lines in our experiment section: “A potential reason for the improved diversity [of FrameFlow]  is the use of SE(3) flow matching in the unconditional model whereas TDS uses SE(3) diffusion (Yim et al., 2023b). Bose et al. (2023) found SE(3) flow matching to provide far better designability and diversity than its diffusion counterpart. Empirically, it is known flow matching outperforms diffusion on Riemannian manifolds (Chen & Lipman, 2023)... A
> > potential reason for RFdiffusion’s overall lower diversity is due to its lack of secondary structure diversity
> > – favoring to sample mostly helical structures.”
> >
> > # References
> >
> > [1] Yim, Jason, et al. "Fast protein backbone generation with SE (3) flow matching." arXiv preprint arXiv:2310.05297 (2023).
> >
> > [2] Chen, Ricky TQ, et al. "Neural ordinary differential equations." Advances in neural information processing systems 31 (2018).
> >
> > [3] Huguet, Guillaume, et al. "Sequence-Augmented SE (3)-Flow Matching For Conditional Protein Backbone Generation." arXiv preprint arXiv:2405.20313 (2024).
> >
> > [4] Lin, Yeqing, et al. "Out of Many, One: Designing and Scaffolding Proteins at the Scale of the Structural Universe with Genie 2." arXiv preprint arXiv:2405.15489 (2024).
> >
> > [5] Wang, Jue, et al. "Scaffolding protein functional sites using deep learning." Science 377.6604 (2022): 387-394.
> >
> > [6] Alamdari, Sarah, et al. "Protein generation with evolutionary diffusion: sequence is all you need." bioRxiv (2023): 2023-09.

---

### Review · Reviewer_awUG · 2024-06-19

**Summary Of Contributions:**

The authors propose additional methods to generate protein structures conditioned on motifs. They compare new approaches as well as additional data augmentation techniques.

**Audience:**

Yes

**Claims And Evidence:**

Yes

**Requested Changes:**

I think Figure 1 could be made even more clear by making all the atoms of the Scaffold red (or warm tones) and all the atoms of the Motif blue (or cool tones). While I like the concept of the figure, it is slightly confusing to me that there are blue atoms on the red Scaffold residues.

Typo: “We use data augmentation to amortize over all possible motifs in our traiing set and aid in generalization to new motifs.”

It would be interesting to discuss the impact of multimers on the proposed methods.

See Weakness section about Figure 3. It needs to be made much clearer to highlight contributions.

In the Diversity analysis section, each of these generated structures is equally likely to fold? How do you ensure that the generated structures are actually valid?

**Strengths And Weaknesses:**

Strengths

I think the approach is broadly good and the augmentation approach is clever.

I think the math in applying the approach to rotations and translations is well defined.

I think Figure 5 is an interesting observation in the generation bias between FrameFlow and RFdiffusion. Just curious–what does the distribution of the training set look like?

Weaknesses

How realistic is the Motif data augmentation approach? Are annotated motifs close in 3D space? In your sampling algorithm, does the distribution of motif residue distances match the distribution of a known dataset?

Figure 3 is very difficult to interpret. Can the Y axis be labeled? Can this data be summarized in some way? If you don’t see any clear trend, why show it? What data do you have to suggest that RF diffusion has the highest success rate–can this be backed up by a statistic, like a p-value?

Neutral

The authors note their method increases diversity of generated structures. Is it explicitly defined in any objective function? What is the mechanism for this increase in diversity? Is this controllable in some way itself?

---

> ### Author Response · Authors · 2024-06-29
> **Rebuttal (part 1)**
>
> We thank the reviewer for their constructive feedback and time. We are glad the reviewer acknowledged the strengths of our approach. Below we address each comment line by line.
>
> > I think Figure 5 is an interesting observation in the generation bias between FrameFlow and RFdiffusion. Just curious–what does the distribution of the training set look like?
>
> Figure 5 demonstrates FrameFlow is able to capture a wider secondary structure diversity of scaffolds than RFdiffusion. We have calculated the same secondary structure composition plot for the training set and have included it in the updated Figure 5. As we can see, FrameFlow’s distribution of scaffolds is closer to the training set than RFdiffusion. Note the secondary structure distribution over the designable scaffolds for the 24 motifs is not expected to exactly match the distribution over the training set (since the training set is over 20,0000 structures).
>
> > How realistic is the Motif data augmentation approach? Are annotated motifs close in 3D space?
>
> Our data augmentation approach trains over all possible motifs in our training set since we randomly sample motifs from each protein structure. Our approach is realistic in the sense that for each training motif we know the true scaffold (which is the remainder of the protein).
>
> We argue this is a sensible approach since it is impossible to know future motif-scaffolding scenarios in advance. Certain tasks may require the motifs to be close in 3D space while other tasks may put them far apart. The lengths and number of motifs may also differ between tasks hence we sample the lengths and numbers during augmentation. Hence, our data augmentation approach is amortizing over all possible motif-scaffolding scenarios in our training set with the hope that this generalizes to unseen motifs.
>
> > In your sampling algorithm, does the distribution of motif residue distances match the distribution of a known dataset?
>
> As previously described, our approach attempts to capture all possible scenarios for motif-scaffolding. That said, our evaluation is based on an unseen test set of previously solved motif-scaffolding problems introduced in RFdiffusion. We argue the performance on the benchmark is indicative of how well our data augmentation approach transfers to realistic scenarios which we find it does.
>
> > Figure 3 is very difficult to interpret. Can the Y axis be labeled? Can this data be summarized in some way? If you don’t see any clear trend, why show it? What data do you have to suggest that RF diffusion has the highest success rate–can this be backed up by a statistic, like a p-value?
>
> Thank you for the suggestions. We have updated figure 3 with a labeled Y-axis in the latest manuscript and have updated the caption with a summary of the figure: “Top plot: RFdiffusion achieves the most designable scaffolds amongst all methods in 9/24 test motifs compared to FrameFlow-amortization’s 7/24 and TDS’ 6/24; 2/24 are ties. Bottom plot: However, we observe that RFdiffusion produces the highest number of unique designable scaffolds for only 2 out of the 24 test motifs. Therefore, previous approaches that only measure designability (top plot) may be misleading since those generative models that may have the best designability can also be repeatedly sampling similar scaffolds. This demonstrates the need to measure diversity alongside designability and use the number of unique designable scaffolds as the metric of success.”
>
> Our purpose is to emphasize that previous methods only measured the number of designable scaffolds which can be misleading for comparing different methods since the model might repeatedly sample highly similar scaffolds.  We estimate sample ‘clusters’ so as to remove samples which are very similar, leading to the ‘diversity’ metric. We use Figure 3 to show this trend of methods like RFdiffusion achieving high success but low diversity. We have added quantities such as the number of motifs RFdiffusion achieves better designability than FrameFlow and the number of motifs RFdiffusion does not achieve better diversity.

---

> > ### Author Response · Authors · 2024-06-29
> > **Rebuttal (part 2)**
> >
> > > The authors note their method increases diversity of generated structures. Is it explicitly defined in any objective function? What is the mechanism for this increase in diversity?
> >
> > A generative model should generate structures with the high level of diversity seen in the training dataset. We believe our approach is closer to an ideal generative model since our objective follows the flow matching generative modeling loss (with a physical auxiliary loss for timesteps close to data) while RFdiffusion utilizes a wide range of auxiliary losses and uses an approximation to the score matching. The RFdiffusion objective may therefore be less well aligned with trying to learn a true generative model. A thorough investigation of the causes of lower diversity in RFdiffusion is not in the scope of our work but previous works have found RFdiffusion to favor alpha helices in its samples which may also explain the lower diversity [1]. As Figure 5 shows, FrameFlow is able to sample a wider distribution of secondary structures which contributes to the increased diversity.
> >
> > > Is this controllable in some way itself?
> >
> > This is a great question and something we are exploring in a follow-up work. We believe it is a natural question to start injecting classifier(-free) guidance techniques to control properties of the scaffolds. We believe secondary structure conditioning should be possible and would control diversity but have not made major progress on this. We have added this possibility as future work in our conclusion.
> >
> > > I think Figure 1 could be made even more clear by making all the atoms of the Scaffold red (or warm tones) and all the atoms of the Motif blue (or cool tones). While I like the concept of the figure, it is slightly confusing to me that there are blue atoms on the red Scaffold residues.
> >
> > Thank you for this suggestion. We have modified figure 1 and 2 to color all scaffold atoms as red and motif atoms as blue. Note we lost the original pymol session files so had to choose new samples to use in the figure.
> >
> > > Typo: “We use data augmentation to amortize over all possible motifs in our traiing set and aid in generalization to new motifs.”
> >
> > Fixed. Thank you!
> >
> > > It would be interesting to discuss the impact of multimers on the proposed methods.
> >
> > Indeed, this was implied in our conclusion “further extensions include binder, enzyme, and symmetric design.” Modeling multimeric proteins would enable extending the methods proposed in our work to binder, enzyme, and symmetry design. We have clarified that multimeric modeling is a future direction in our conclusion.
> >
> > > See Weakness section about Figure 3. It needs to be made much clearer to highlight contributions.
> >
> > Following our previous comment, we have updated the caption and included more discussion about Figure 3 in the updated manuscript.
> >
> > > In the Diversity analysis section, each of these generated structures is equally likely to fold? How do you ensure that the generated structures are actually valid?
> >
> > Yes, we only perform our diversity analysis on generated structures that pass the designability criterion – which has the interpretation of “likely to fold from a protein sequence”. As described in Appendix E, we use the “designability” criterion to determine if generated structures are valid. This metric was established in RFdiffusion to be a good predictor of valid structures. We note this metric is not perfect since it relies on using two black box neural networks, ProteinMPNN and AlphaFold2, to verify our structures. Despite its limitations, this metric is widely used in the current literature [2].
> >
> > # References
> >
> > [1] Huguet, Guillaume, et al. "Sequence-Augmented SE (3)-Flow Matching For Conditional Protein Backbone Generation." arXiv preprint arXiv:2405.20313 (2024).
> >
> > [2] Bennett, Nathaniel R., et al. "Improving de novo protein binder design with deep learning." Nature Communications 14.1 (2023): 2625.

---

### Author Response · Authors · 2024-06-30
**Global response**

We thank the action editor and all the reviewers for their constructive feedback and time. We are glad all reviewers attributed clarity and novelty as a strength of our work. We have responded to each reviewer line by line. Our manuscript has also been revised with updated text indicated in red. Our figures have also been improved based on feedback from reviewers.

---

### Decision · Action_Editor_HowJ · 2024-07-15

**Recommendation:** Accept as is

**Comment:**

All Reviewers unanimously supported accepting the paper. Among the strengths of the paper, Reviewers appreciated clarity of writing, comprehensive experimental setup, and the additional evlauation of the diversity of proposed scaffolds. Authors engaged in the rebuttal and addressed major comments.

Thank you for your submission and it is my pleasure to recommend accepting the work.

I would like to make a minor suggestion to the work. I think readers would benefit from providing a motivation in the introduction why modifying FrameFlow is the natural way to increase diveristy of the proposed solutions. One may think that modifying other methods might or might not also bring benefit, so some elaboration on this topic could be useful. It would also seem to me that the technical introduction to CNFs might be too dense for certain audiences. Perhaps there is a way to expand it a bit and make it more intuitive to less mathematical audience?

**Audience:**

The paper will be of high interest to the community interested in motif-scaffolding of proteins. Aspects of the overall technique can likely be inspiring for other molecular and protein design tasks.

**Claims And Evidence:**

The core claim is that the introduced variant of FrameFlow achieves the state of the art in terms of the diversity (over 2x improvement) of the generated scaffolds. As all Reviewers agree, the claim are clearly supported by the provided experimental data.